**A new accurate low-cost instrument for fast synchronized spatial measurements of light spectra**

Bert G. Heusinkveld, Wouter B. Mol, Chiel C. van Heerwaarden

*Meteorology and Air Quality Group, Wageningen University & Research, P.O. Box 47, 6700 AA Wageningen, The Netherlands*

**Abstract**

We developed a cost-effective Fast Response Optical Spectroscopy Time synchronized instrument (FROST). FROST can measure 18 light spectra in 18 wavebands ranging from 400 to 950 nm with a 20 nm full width half maximum bandwidth. The FROST 10 Hz measurement frequency is time-synchronized by a Global Navigation Satellite System (GNSS) timing pulse and therefore multiple instruments can be deployed to measure spatial variation of solar radiation in perfect synchronization. We show that FROST is capable of measuring global horizontal irradiance (GHI) despite its limited spectral range.

It is very capable of measuring Photosynthetic Active Radiation (PAR) because 11 of its 18 wavebands are situated within the 400 to 700 nm range. A digital filter can be applied to these 11 wavebands to derive the Photosynthetic Photon Flux Density (PPFD) and retain information of the spectral composition of PAR radiation.

The 940 nm waveband can be used to derive information about atmospheric moisture.

We showed that the silicon sensor has undetectable zero offsets for solar irradiance settings and that the temperature dependency as tested in an oven between 15°C and 46°C appears very low (-250 ppm K$^{-1}$). For solar irradiance applications, the main uncertainty is caused by our Poly Tetra Fluor Ethylene (PTFE) diffuser (Teflon), a common type of diffuser material for cosine-corrected spectral measurements. The oven experiments showed a significant jump in PTFE transmission of 2% when increasing its temperature beyond 21°C.

The FROST total cost (<€200) is much lower than current field spectroradiometers, PAR sensors or Pyranometers, and includes a mounting tripod, solar power supply, datalogger and GNSS and waterproof housing. The FROST is a fully stand-alone measurement solution. It can be deployed anywhere with its own power supply and can be installed in vertical in-canopy profiles as well. This low cost makes it feasible to study spatial variation of solar irradiance using large grid high-density sensor set-ups or to use FROST to replace existing PAR sensor for detailed spectral information.

**1. Introduction**

Understanding solar irradiance and its interaction with clouds and vegetation is of utmost importance to unravel the complexity of feedback systems that determine our weather and climate. Cloud-shading dynamics of irradiance are highly dynamic (Lohmann, 2018) and Cloud-Resolving Models (CRM) are unable to resolve short time intervals and small spatial scales. At grid scales below 1 km, 3-D radiative transfer models can greatly improve the 3-D surface and atmosphere heating rates in atmospheric models (Calahan et al., 2005, Jakub & Mayer, 2015). A good example is the complexity of the radiative effects of shallow cumulus clouds and its interactions with a vegetated surface. Traditional 1-D radiation models produce unrealistic surface radiation fields but Menno et al. (2020) showed that a 3-D radiation transfer model could greatly improve the coupling mechanisms between clouds and the land surface. The small circulations, turbulence and combined cloud microphysics in convective boundary layers are both highly non-linear and complex. CRMs are crucial for improving weather forecasting models and for the energy meteorology sector. Kreuwel et al. (2020) showed that solar powered grid loading is highly dynamic and especially so for smaller household PV systems, leading to grid overload challenges at very short time intervals of seconds. High quality observations, both in high resolution spatially and with a high temporal resolution, are required to test such models but so far observations are lacking (Guichard and Couvreux, 2017).

Yordanov et al. (2013) showed that cloud enhancements can significantly increase solar irradiance levels (>1.5 times), which result in peak irradiance levels well exceeding extraterrestrial levels even at high altitudes and latitudes (Yordanov, 2015). They used fast response silicon sensors and their highest detected irradiance bursts lasted about 1 s, which led them to believe that the required light sensor response time should be at least 0.15 s, much faster than traditional thermopile pyranometers with a response time of several seconds. The slow response time of those thermopile sensors is related to the thermal mass of the thermopile sensor. Semiconductor light sensors respond faster because photons directly mobilize electrons that can be measured directly. The downside of semiconductor light sensors is their limited and non-flat spectral response and temperature sensitivity. Thermopile based pyranometers are also expensive as compared to a silicon-based solution, which limits their large-scale use in meteorological measurement networks. Martinez et al. (2009) showed that a factor of 10 reduction in pyranometer costs as compared to a thermopile sensor is possible with the use of a silicon photodiode, however their spectral response is limited (400 to 750 nm) and it has non-flat spectral response. A major solar spectral change occurs in the infrared due to water absorptions bands, which leads to an overestimation for clear sky conditions and an underestimation for overcast skies when calibrated for average weather conditions.

The spectral response limitations of the photodiode used by Martinez et al. (2009) can be improved with a wider spectral response silicon type pyranometer such as applied in the LI-COR 200-SZ as demonstrated by Michalsky et al. (1991). They compared the LI-COR 200-SZ with a thermopile pyranometer (Kipp & Zonen CM-11). The CM-11 has a flat spectral response (300 to 2500 nm) whereas the LI-COR 200-SZ exhibits a very nonlinear and limited spectral response starting at 400 nm and increasing 5-fold in sensitivity towards its peak around 1000 nm, then sharply dropping off to zero at 1100 nm (Alados-Arboleda et al., 1995). Their main uncertainty is related to the temperature dependance of silicon sensors. After a temperature correction, they performed similarly to thermopile pyranometers (11.4 W m$^{-2}$ rms errors) under clear and cloudy sky conditions. This is surprisingly accurate because LI-COR calibrates their pyranometer against a reference thermopile pyranometer and therefore a change in solar spectrum may affect its accuracy. Michalsky et al. (1990) argued that the clear or cloudy sky global horizontal irradiance (GHI) spectra is similar because of clouds mixing the direct and blue skylight. This, however, is contradicted by a recent study by Durand et al. (2021) where they investigated the spectral differences between clear and overcast skies. They showed that clouds, in relative terms, enrich GHI spectra in wavelengths < 465 nm and is depleted in wavelengths > 465 nm. This may well explain why the LI-COR sensor performed so well because its main sensitivity is in wavelengths > 465 nm thus indirectly correcting for the reduced infrared in the major water absorption bands beyond its spectral range.

Optoelectronics are evolving rapidly and innovations in semiconductor integration with optical components and microprocessors are paving the way for cost-effective spectrometers that can provide even temperature-compensated spectral details about solar radiation. A leading manufacturer in this field is AMS (Austria Micro Systems, Austria) and offers various intelligent light sensing products that are capable of measuring light intensity within multiple optical wavebands. These sensors are mass-produced, resulting in low-cost sensors. Tran and Fukuzawa (2020), tested such a cost-effective 18 band multispectral sensor (AS7265x, AMS) for spectroscopy of fruit (between 400 and 950 nm) and useful information could be derived. Such filter spectroscopy sensors would be very interesting for solar global horizontal irradiance measurements (GHI). The spectral signature of radiation is very relevant to quantify since clouds and air pollution modify the solar light spectrum and light scattering. Additionally, multiple reflections between various ground and water surfaces and clouds will further influence the light spectral composition. This is especially relevant in the photosynthetic active radiation wavelengths (PAR) for vegetation cloud feedbacks since it affects photosynthesis and evapotranspiration (Durand et al., 2021).

Here we present the development of a cost-effective fast-response solar light sensor grid for spatially and temporally high resolution multiple light waveband-resolved GHI measurements. The required large number of sensors requires cost-effective design optimization.

Additionally, we tested these sensors for meteorological, photosynthesis and remote sensing applications
as well as performance both in the lab and in field experiments.
**2. Instrument design and measurement method**
The measurement system we developed is depicted in Fig. 1 and consists of a silicon light sensor chipset
(AMS AS7265x), a GNSS for time synchronization, a cosine corrector light diffusing input port, and a
microcomputer. See Table 1 for a list of components.

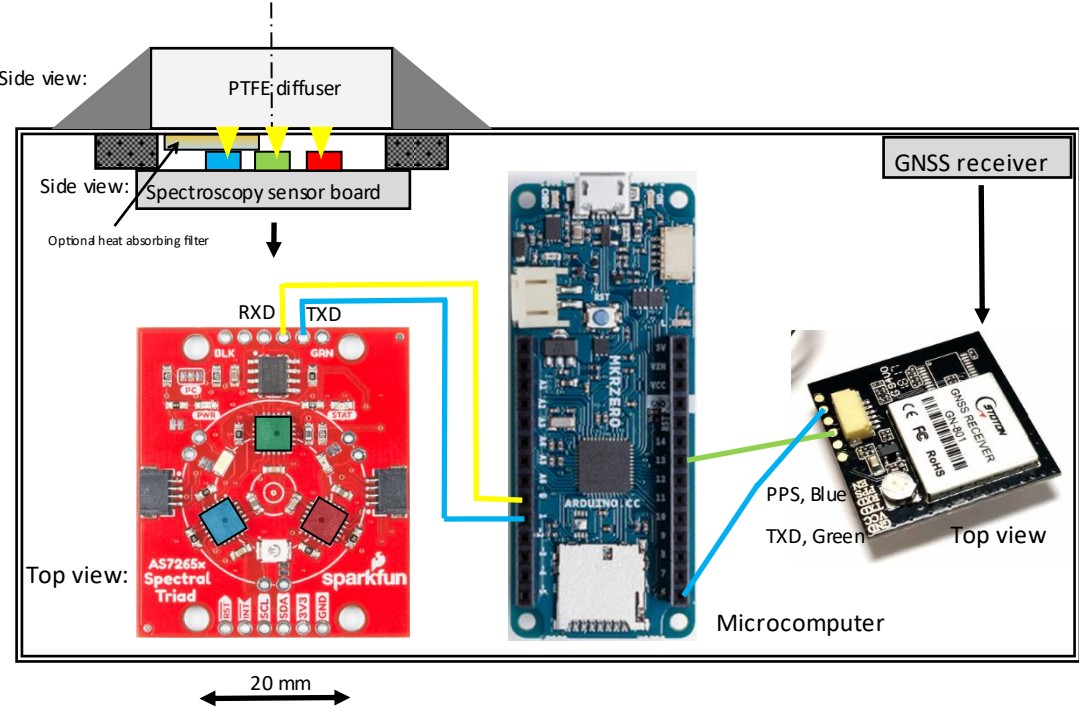


Figure 1: Mechanical layout and wiring diagram of the FROST spectrometer (Sensor: 3.2 mm below a
Teflon filter with radius of 32 mm). For easy identification we color-coded the three light sensors (blue,
green and red), each measuring 6 channels.
Time-synchronized measurements are achieved using a hardware GNSS receiver timing pulse (PPS) to
trigger each measurement and time-stamped data is processed and collected by a microcomputer board
(Fig. 1, Table 1).
Table 1: List of components for the waterproof solar powered spectrometer.

| Component | Manufacturer and model | Price (€) |
|---|---|---|
| Spectroscopy chipset | AS7265x spectral sensors triple AMS (Austria) with interfacing logic mounted on a PCB by Sparkfun (U.S.A.) | 70.00 |
| Optical filter | Schott heat-absorbing colored glass filter KG3 or KG1, 2 mm (Germany) | 11.00 |
| UV-sensor | GUVA-S12SD (optional, with a thin second PTFE diffuser) | 1.70 |
| PTFE diffuser | 32 mm diameter, cut from a plate (S-Polytec GmbH, Germany) | 3.00 |
|  |  |  |
| GNSS receiver | TOPGNSS GN-901, China, GPS and Glonas receiver | 6.00 |
| Microcomputer | Arduino MKR Zero | 23.00 |
| Memory card | Kingston Canvas Select Plus microSDHC 32GB | 4.00 |
| Breadboard | Solderless PCB breadboard Mini protoboard | 0.90 |
| Solar panel | First Solar, China, CNC165x165-5, Polycrystalline, 4.2 W, 5 V, 840 mA, 165x165 mm | 7.00 |
| Battery | Li-Ion battery LP906090JH, Jauch, Germany | 30.00 |
| Charger controller | Mini Solar Lipo Charger board CN3065 | 1.40 |

| | | |
|---|---|---|
| **Box** | **Outdoor Junction Box 100x150x70mm waterproof IP65, Shockproof ABS plastic. ManHua, China (AliExpress)** | **9.00** |
| **Tripod mount adapter** | **Camera metal shoe mount adapter ¼" thread** | **0.75** |
| **Tripod** | **König KN-TRIPOD21/4 camera tripod pan & tilt 130 cm** | **12.00** |
| **Ground anker** | **Tent herring** | **1.00** |
| **Silicone adhesive sealant** | **Permatex 81158 or Bizon Black Silicone Adhesive** | **1.00** |


The light sensors are mounted on camera tripods, which makes leveling easy (Fig. 2). A camera metal
shoe mount adapter was glued under the polycarbonate housing for fast mounting. For winds >6 m/s it
is advised to use tent herrings to fix the tripod to the ground. The power consumption is 0.5 W and a 6
Ah LiPo battery will last for 40 h without sunshine. Battery capacity is reduced at lower temperatures.
The 4.2 W polycrystalline solar panel together with a LiPo charge regulator is a reliable power supply
solution for continuous operation in the Dutch climate from April - September. The solar panel is glued
on a special shaped wooden frame that slides over the tripod center tube, with the solar panel sides
resting against the two outer tripod leg (Fig. 2). It is fixed to the rear leg with a thin metal wire. Hot glue
appeared unsuitable for the panels and it is advised to use epoxy glue.
The PTFE diffuser was glued to the box by roughening the surfaces and using a black silicone adhesive
around the diffuser edges.

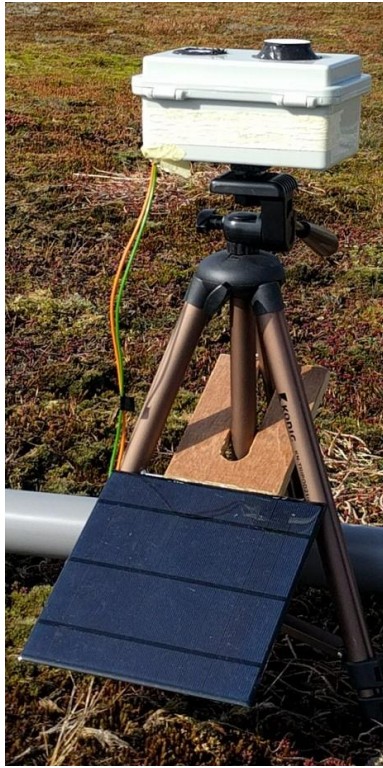


Figure 2: FROST with solar panel mounted on a camera tripod.
The correct synchronization of the sensor grid measurements is essential and several options were
considered such as a network configuration with synchronized triggering at fixed time intervals. Wires in
the field were not an option due to logistic challenges, and radio communication could be possible but
adds to the cost with reduced reliability due to radio interference. As a robust option, a GNSS receiver
was chosen that constantly synchronizes its internal time to an international clock standard. Similar
timing synchronizations are used for sensors grids in seismic activity monitoring of volcanos where
timing is essential to determine seismic propagation and where synchronization accuracy of 50 ns could
be achieved (Lopez Pereira et al., 2014).

**2.1 Light sensor**
The light sensing element is the AMS AS7265x, a smart spectrometer sensor capable of measuring light
at 18-Channel 20 nm full width half maximum (FWHM) bandwidth from visible and near infrared spectral
bands (410 to 940 nm) with an electronic shutter (manufacturer: AMS, Australia). It has a broad
operational temperature range from -40°C to 85°C. The spectrometer consists of three separate
integrated circuits with each including six silicon-based photo diodes with integrated optical bandpass
interference filters, micro-lenses, a programmable analog amplifier, and an analog to digital converter
and a microprocessor. We will identify the AS72651, -52, -53 as the blue, red and green sensor, as
indicated in Fig. 1. The integrated light interference filters are directly deposited on the silicon. Factory
calibration values are stored inside the internal memory. Two serial communication options are available
for interfacing with a microcomputer; a Universal Asynchronous Transmission (UART) and a synchronous
serial transmission (I2C) port. The three light sensor view angles are limited by the chip housing light
input port to 41°, which ensures that the optical interference band filters stay within the 20 nm FWHM
and +/- 10 nm center-wavelength specifications. AMS states that their filter stability (in time and against
temperature) is not detectable but does not provide further specifications. They do mention that the
wavelength accuracy is within +/- 10 nm. The AS7265x triple set of light sensor chips, each capturing six
light wavebands, poses a challenge to couple optically all three to the same sensing area and to assure a
good cosine response needed for the accurate measurement of GHI. The limited opening angle poses an
additional challenge for GHI measurements since they require a viewing angle of 180°, therefore an
achromatic cosine-corrected diffuser is required.
**2.2 Diffuser material**
Teflon (PTFE) material is commonly used as an effective light diffuser, with a large spectral transmission
range starting below 300 nm, and is available from various manufacturers. However, PTFE light
transmittance exhibits a temperature dependency caused by a major phase change in its crystalline
structure at 19°C. The phase change can cause a significant change in transmittance. Yliantilla and
Schreder (2005), tested three commercially available PTFE diffusers and found transmission changes
between 1% and 4% at the phase change temperature. By comparison, they also showed a quartz
diffuser with a linear response to temperature (0.035% °C$^{-1}$) without the sudden transmission jump as
found in the PTFE diffusers. Despite this, PTFE was nevertheless chosen as a cost-effective diffuser to
maximize the spatial number of sensors. The diffusers were cut from PTFE plates (S-Polytec GmbH,
Goch, Germany) using a vice and a hole punch to press round diffusers. A 10.6 and 2.0 mm thick diffuser
were tested. The transmission temperature dependency of our PTFE diffusers was tested in a
temperature-controlled oven with a cooler (WTS Binder, Germany with a EUROTHERM temperature
controller). The oven is equipped with a front glass door and the lowest possible temperature setting was
kept above the dewpoint temperature of the laboratory to avoid moisture condensation issues. An LED
light source (LCS, 17 W, 2500 Lumen) was chosen for its high output and limited thermal infrared and
powered by a stabilized voltage power supply. The LED was placed outside the oven in front of the oven's
glass door about 1 m away to minimize lamp heating. A second light sensor was placed outside the oven
next to the lamp to monitor its output. Diffuser light transmission measurements were corrected for
variation in lamp output. Subsequently the light sensor without a diffuser was tested. Temperature
sensitivity measurement results are presented in Section 3.2.
The spectrometer performance was also tested at the DWD (German Weather Service) radiation
calibration facility in Lindenberg, Germany. The spectrometer output was compared against a calibrated
xenon light source and the intensity was adjusted by varying the lamp sensor distance between 0.5 m
and 0.7 m. The possible spectral crosstalk of infrared light was tested by placing a very steep long pass
interference filter with a Cut-On Wavelength of 1000 +/-9 nm (Dielectric Coated Long pass Filter, 25.4
mm diameter, 1.1 mm thick, transmission >95%, OD5 Blocking, Edmund optics, Stock #15-463) in front
of the sensor. The long pass filter blocks all sensor wavebands and any remaining signal is then
considered infrared crosstalk. The position of the optical waveband filters was tested using a Cary 4000
(Agilent, USA) UV-Vis NIR spectrophotometer at Wageningen University, The Netherlands. Results are
presented in Section 3.1.
**2.3 Cosine response**
The cosine response was determined by placing a LED light source (LED light bulb 2500 Lumen, diameter
0.1 m) and our light sensor 5 m apart, both on tripods at 1 m height. Since a darkroom was not
available, the measurements were performed outdoors at night to avoid reflection from ceilings and
walls. A night with low humidity was chosen to minimize aerosol light scattering. The direction of the
light sensor was adjusted from 0° (viewing the light source) to 90° (perpendicular to the light beam). To
keep the distance between the sensor and light source constant during rotation, the plane of rotation was
exactly located at the diffuser surface. A shading screen was placed between the light source and sensor
to shade the ground surface to avoid any light reflection into the sensor. Results are presented in Section
194     3.3.

**2.4 Time synchronization**
Instead of using a GNSS for synchronizing an internal clock or using the serial date and time output, we
use the very precise (<100 ns accuracy) hardware timing pulse of a GNSS module to trigger each
measurement directly (at 10 Hz.). The data is time-stamped with the GNSS date and time output. A
special GNSS receiver was selected that also outputs a programmable timing pulse for synchronization
purposes (better than 50 ns). As a bonus, it also provides location data within a few meters. These
receivers can be purchased for less than 6€ (Table 1). The time synchronization analysis can be found in
Section 3.4.
**2.5 Datalogging**
The datalogging of the GNSS date, time, latitude, longitude and the 18 channel spectroscopy
measurements at 10 Hz results in a dataflow of >100 MB data per day. The spectroscopy sensor outputs
ASCII data and the bandwidth of the I2C interface on the spectroscopy side were insufficient, thus the
UART serial interface was selected. The sensor can be triggered by serial command to do a measurement
and this command in turn is triggered by the hardware timing pulse of the GNSS.
For datalogging, the MKR Zero of the Arduino family microprocessor platforms was chosen. It is a cost-
effective and low power datalogging solution using a 48 MHz SAMD21 Cortex 32bit low power ARM MCU
and a built-in micro-SD card holder (max. 32 GB). A consumer grade 32 GB SD card was selected, data
rates are low (<5 KB s$^{-1}$) and the large size ensures that the card does not wear down fast (<4 GB
month$^{-1}$). The challenge with this datalogging solution is that the default operating system cannot handle
sustained data writing to an SD card at 10 Hz using linear programming (despite a low data rate of <5
KB s$^{-1}$). In fact, the SD card would regularly delay the measurements with an estimated 200 ms resulting
in a loss of data (tested with a new, fast SD card with 85 MB/s max write speed). Thus it needs a
microcontroller multitasking real-time operating system and FreeRTOS (freertos.org) was chosen to
overcome this. Two tasks that run semi-parallel on the single core CPU were defined. The first task with
the highest priority will initiate a measurement cycle at the falling edge of the hardware timing signal of
the GNSS. Task 2 will trigger each second and writes the 10 Hz buffered data to the SD card (Fig. 3).

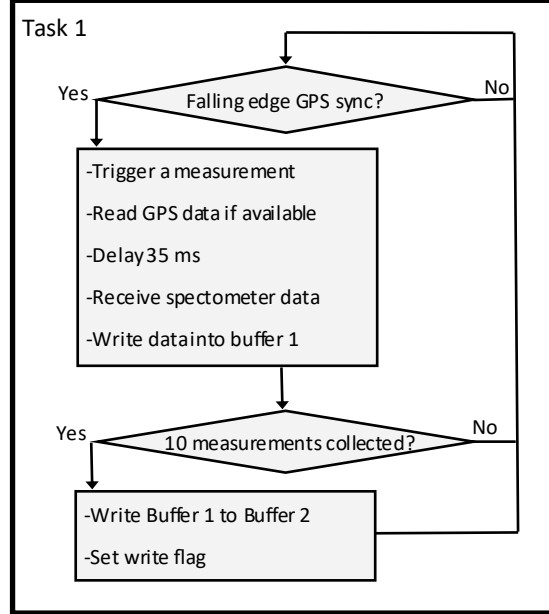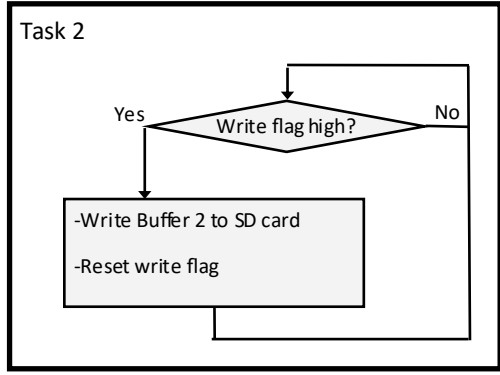


Figure 3: Multitasking software implementation for synchronized measurements and data storage. Each Buffer can contain 10 rows of data. The program is available at zenodo.org (DOI 10.5281/zenodo.6945812).

## 2.6 Field experiments

The field experiments were conducted at various locations. At the Veenkampen weather station, Wageningen, The Netherlands (Lat.: 51.981°, Long.: 5.620°), sensor performance was tested against GHI measurements and a spectrophotometer. Although GHI is directly measured with a pyranometer, it was decided to use the pyrheliometer and diffuse radiation sum to reduce cosine response errors. The instruments consist of a Kipp and Zoonen Pyrheliometer CMP1 with a calibration accuracy of +/-0.5% and a first class pyranometer CM21 for diffuse radiation measurements with a time constant of 5 s, directional error <+/-10W $m^2$, tilt error: +/- 0.2%, zero-offset due to T change: <2 W $m^2$ at 5K h-1, cosine response error: max +/-2% at 60°, max. +/-6% at 80°. Both instruments were mounted on a suntracker (EKO instruments, Japan, STR-21 with shading disk). On selected days the solar spectrum was measured with the ASD FieldSpec (U.S.A.) field spectroradiometer with a cosine collector and with a factory recalibration made in 2021.

Additionally, a large set of sensors were deployed during the FESSTVaL campaign (https://fesstval.de/) at the German Weather Service (DWD) in Falkenberg, Germany, to study the spatial variation of solar irradiance (June 2021). For that campaign, it was crucial to obtain fast and time synchronized spatial solar irradiance measurements. Their Baseline Surface Radiation Network (BSRN) location at Lindenberg (Driemel et al., 2018) was used to test long-term stability from 22 June-31 August 2021.

The FROST was also deployed in a field experiment in La Cendrosa, Spain (Lat: 41.692537, Long: 0.931540) from 14-29 July 2021. It was used, among other things, to study crop growth.

## 3. Performance and applications

The performance of the sensor, the temperature dependency and cosine response of the diffuser and the time synchronization are presented below. The infrared crosstalk is analyzed by measuring signal response with all light below 1000 nm blocked using a low-pass infrared filter. Subsequently a correction method using heat absorbing infrared filters (referenced to as "correction filters") is introduced and tested. This results in three versions of FROST: one with a 10.6 mm diffuser, one with a 2 mm diffuser including a correction filter on the blue sensor, and one with a 2 mm diffuser with two correction filters (on the blue and red sensor).

**3.1 Spectral response and calibration**

According to the manufacturer specifications, the normalized (at peak wavelength) responsivity of their spectroradiometer has a good narrow band response (20 nm full width half maximum (FWHM)) and limited overlap for the 18 channels. Wavelength accuracy is within +/- 10 nm and this was confirmed by testing the sensor inside a Cary 4000 UV-Vis spectrophotometer equipped with a universal attachment accessory. Unfortunately, the Cary spectrophotometer had a limited spectral range so we could not test the crosstalk in the near infrared, nor the 940 nm band (Fig. 6a).

Linearity was tested by comparing the spectroradiometer against a reference thermopile pyranometer CM21 (Kipp and Zoonen, The Netherlands) and a stabilized halogen light source in a dark room. The intensity was adjusted by changing the lamp distance. The FROST non-linearity was at least as good as the CM21, which has a non-linearity of <+/-0.2%. The factory-calibrated accuracy is +/- 12% according to the manufacturer specifications (in counts/$\mu$W/cm$^2$). After initial testing using solar radiation as a light source for reflectance measurements of lawn grass, we were confronted with unusual data. The PAR region clearly showed very high reflection values, more than 5 times of what is typical for such a surface.

After consultation with the manufacturer (AMS), they clarified that the AS72653 sensor has a strong crosstalk in the near infrared. The sensor was meant to be used with LED light for spectral reflectance measurement applications, which would not produce light > 1000 nm. They recommend for each sensor a specific LED, and therefore each reflectance measurement would consist of 3 separate measurements with each using one sensor and with one specific LED at a time.

We tested the sensor at the DWD radiation calibration facility in Lindenberg using a calibrated light source. The crosstalk caused by light wavelengths beyond 1000 nm was measured by using an optical Long Pass interference (LP) filter that blocks all light below 1000 nm. Thus, the remaining signal on all 18 channels can be attributed to a crosstalk from wavelengths >1000 nm. The blocking filter characteristics were tested in a Cary 5000 UV-Vis-NIR spectrophotometer equipped with a universal attachment accessory (Fig. 4).

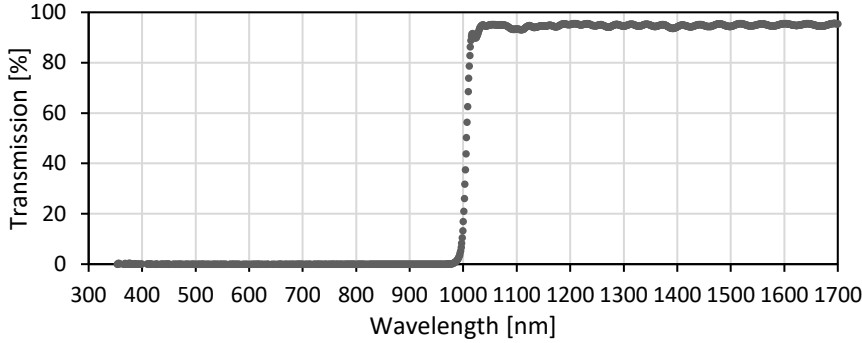

Figure 4: Transmission of the optical LP filter measured with a Cary 5000 UV-Vis-NIR spectrophotometer equipped with a universal measurement accessory at the DWD, Lindenberg, Germany.

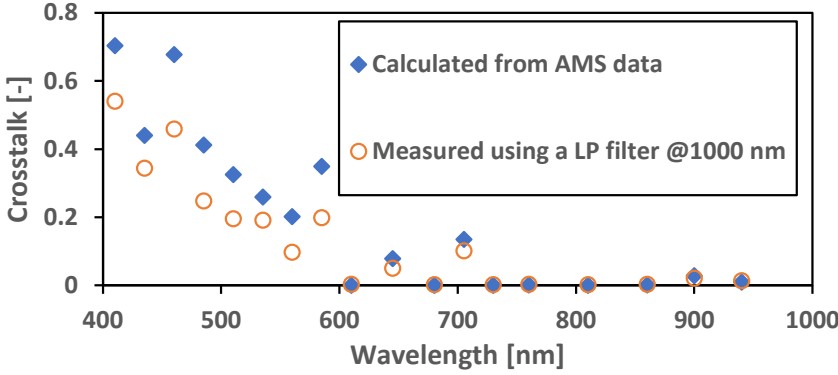

Figure 5: Measured spectroscopy sensor infrared crosstalk from wavelengths >1000 nm, tested with a
Xenon lamp and an optical long pass filter (LP at 1000 nm) and calculated from spectral response data as
supplied by the manufacturer (AMS). Crosstalk is defined as the fraction of light wavelengths >1000 nm
the sensor is responding to, for example at 410 nm 55% of the measured signal is actually originating
from wavelengths >1000 nm.
Figure 5 shows the fraction of infrared light (>1000 nm) within the sensor output for each of the 18
channels. The sensor output was corrected for the LP filter transmission loss (about 5%, see Fig. 4). Note
that the crosstalk is larger than during solar radiation measurements because a Xenon light source
contains a higher amount of infrared radiation. Cloudy conditions would further reduce crosstalk. For
clear sky conditions, the crosstalk would be about half of the Xenon light. The blue dots in Fig. 5 show
the calculated crosstalk using the AMS sensor spectral filter response data obtained through personal
communication with the manufacturer (Fig. 6a). The measured crosstalk on the blue sensor appears to
be slightly better. The crosstalk is very large in the visible light range and confirms the provided filter
transmission curves from AMS (personal communication). Note that those transmission curves (Fig. 6a)
are not available on the publicly available datasheet. Figure 5 shows that only half of the channels
provide the correct spectral information (if calibrated correctly using the data from Fig. 6a). However,
there are enough channels to measure the so-called red edge around 700 nm in vegetation light
transmission and reflection. This opens up applications for vegetation growth measurements without
further modifications.
All channels in the blue sensor and some of the channels (650 and 685 nm) in the red sensor have very
high crosstalk from the 1000 to 1100 nm range, but the crosstalk makes the sensor cover a larger range
of the solar spectrum. It is therefore still usable if this can be quantified.

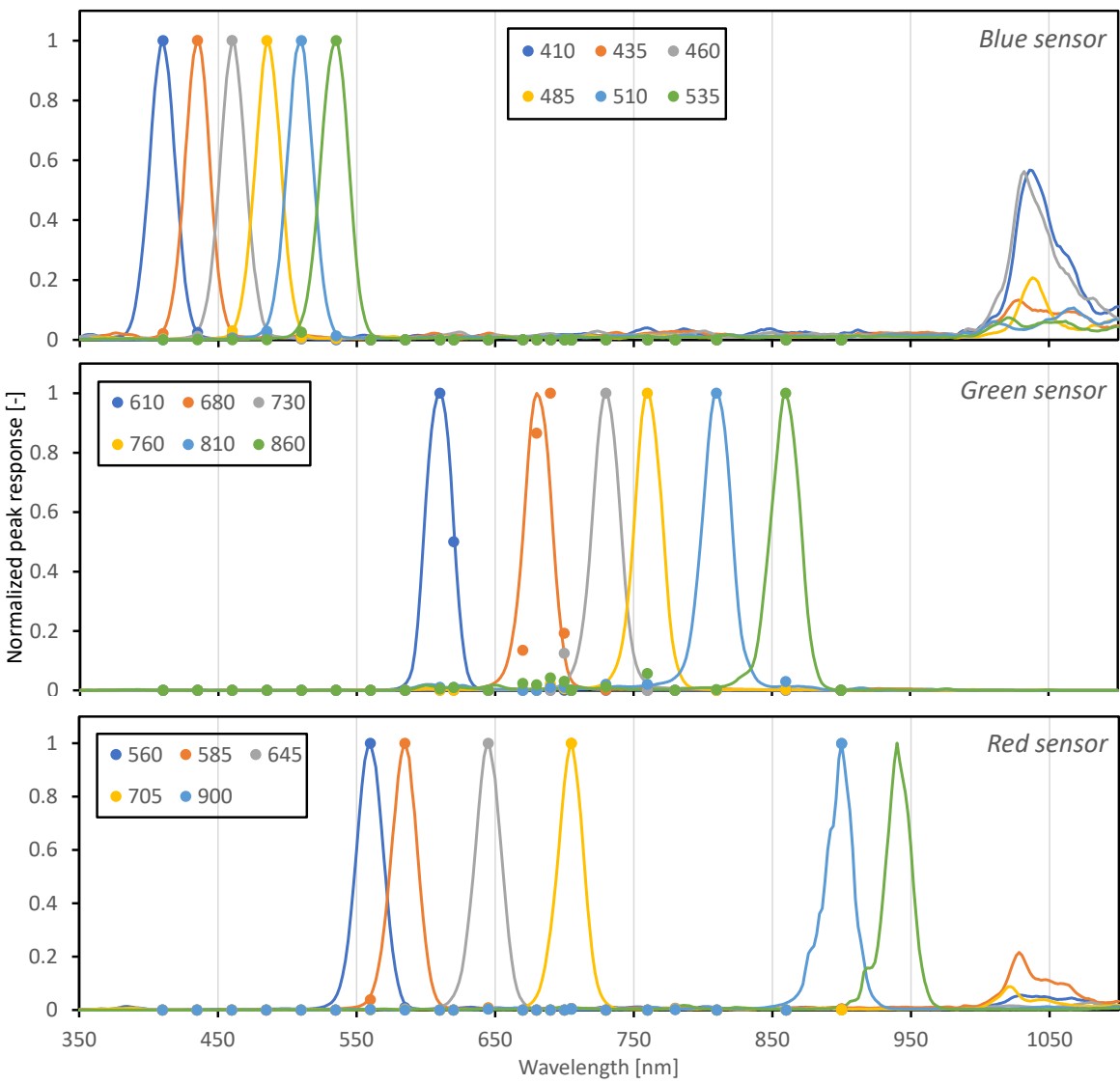


Figure 6a: Normalized peak spectral response of the triple AMS sensor (denoted as blue, green and red
sensor), data provided by manufacturer after consultation (solid lines) and our measured response up to
900 nm (dots) as measured with a sensor placed inside a Cary 4000 UV-Vis spectrophotometer equipped
with a universal measurement accessory.





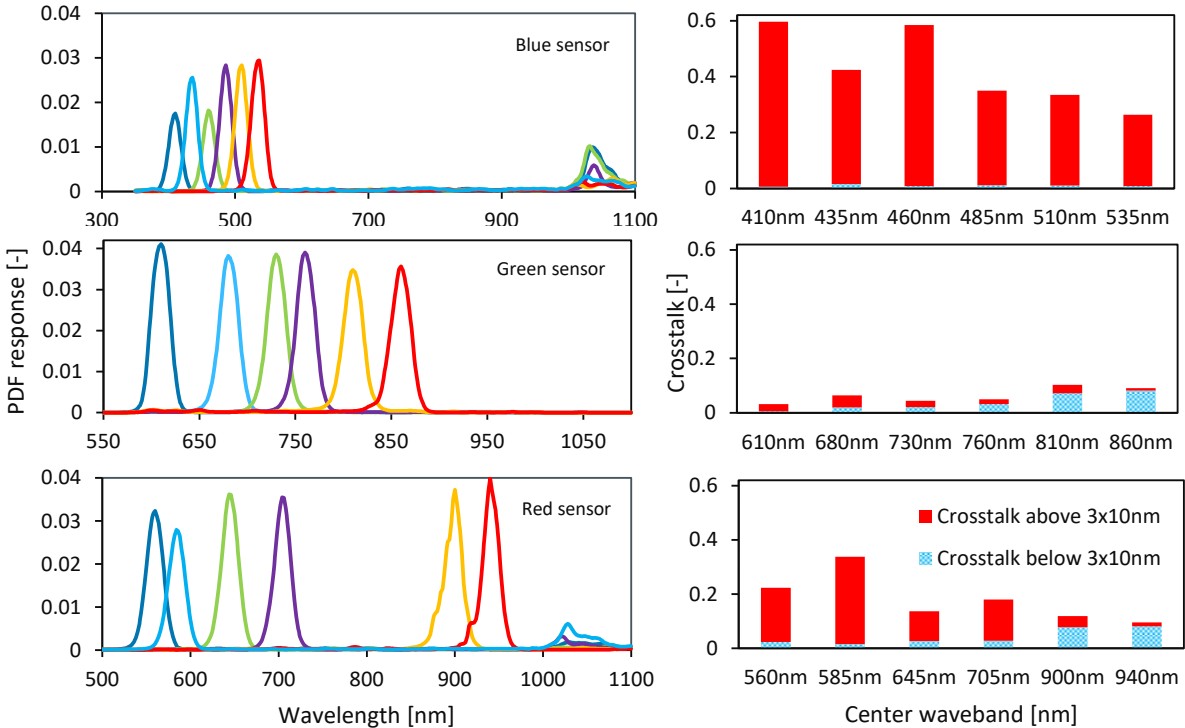


Figure 6b: Left panels: Spectral response of the triple AMS sensor (denoted as blue, green and red sensor), Probability Density Functions (PDF) calculated from data provided by manufacturer AMS (personal communication). Right panels: The crosstalk for each channel is presented as two values: Signal originating from >(center wavelength+30 nm) divided by total signal of a channel (red) and signal <(center wavelength-30 nm) divided by total signal of a channel (blue) (sensor only, without diffuser).

In Fig. 6b, the right panels, show that the crosstalk for a flat spectrum, defined as the signal above or below 3x 0.5 FWHM from the center wavelength, is large (up to 60%) in the PAR range for the blue sensor and mainly from the infrared beyond 1000 nm. The green sensor performs much better and exhibits minimal infrared crosstalk (<5%). The red sensor has an issue mainly with the first two channels.

To remove infrared crosstalk, an optical short pass filter is required. However, a filter with a sharp cut-off at 1000 nm is, to our knowledge, not available or probably very expensive and sensitive to angle of incidence. Cost-effective short pass filters are made from heat-absorbing glass and have a dye added to the glass that absorbs infrared radiation. However, these heat-absorbing filters do not have a steep filter response and therefore ineffective to correct the red sensor without attenuating the 900 and 940 nm channels too much. The Schott heat-absorbing filters KG3 and KG1 appear to offer a good solution for the blue sensor (Fig. 7). The remaining crosstalk is mainly related to the slightly broader filter response. The first 4 channels of the red sensor (Fig. 6b) can also be improved. However, such a correction filter for the red sensor would increase crosstalk from shorter wavelengths for the 900 and 940 nm wavebands (see lower right panel in Figure 6b) and greatly reduce signal strength. For accurate PAR measurements, and when the 900 and 940 nm channel are not needed, it is recommended to use the weaker KG-1 filter for the red sensor.

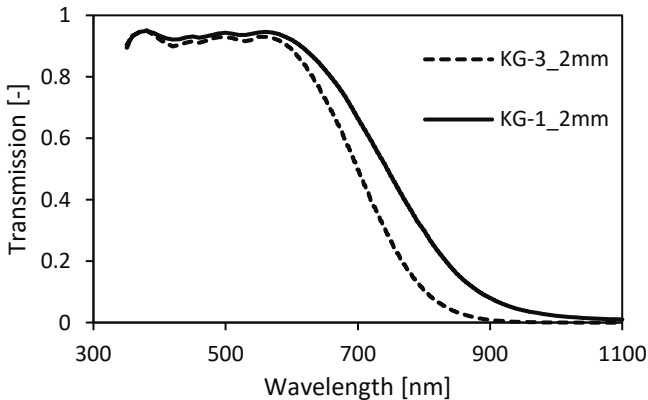


Figure 7: Correction filters for the infrared crosstalk: Schott heat-absorbing filters (adapted from Schott
AG manufacturer data).

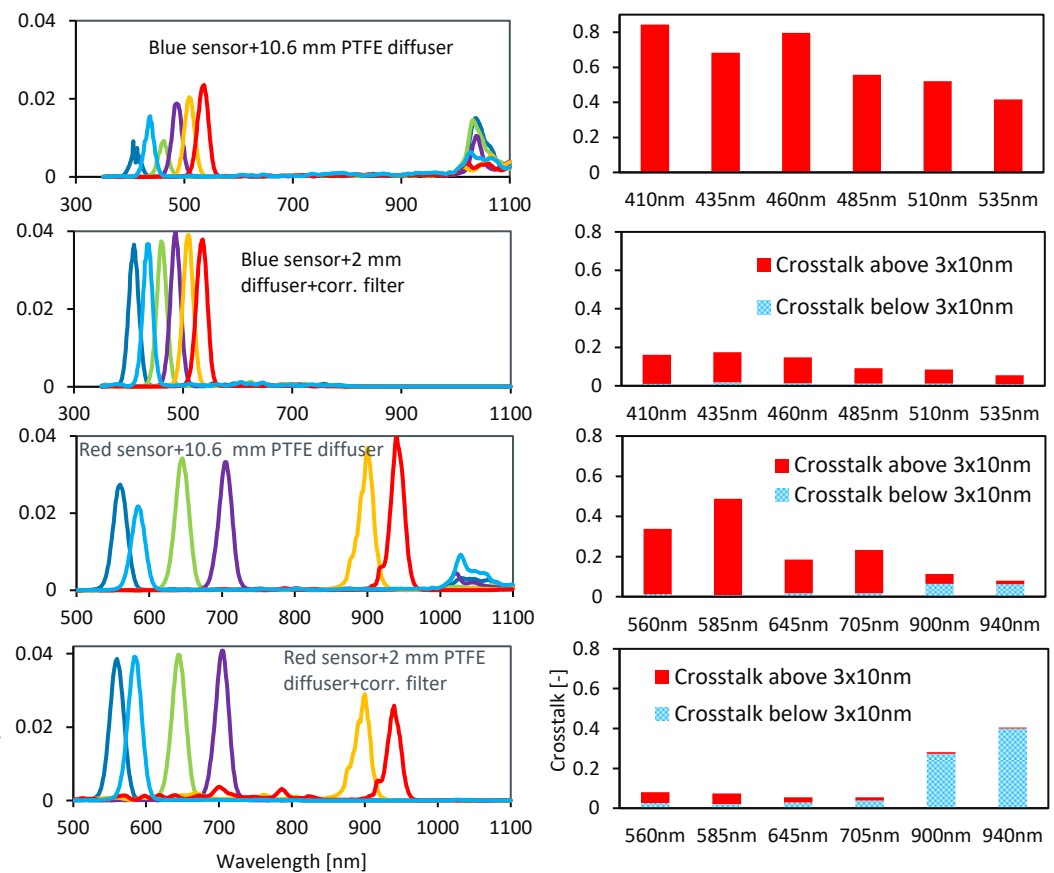


Figure 8: Spectral response and crosstalk (for a flat spectrum) of the blue and red sensor, without or
with correction filter. First row: Blue sensor with 10.6 mm PTFE diffuser. Second row:Blue sensor with 2
mm PTFE diffuser including a heat absorbing filter (Schott KG-3). Third row: Red sensor with 10.6 mm
PTFE diffuser. Fourth row: Red sensor with 2 mm PTFE diffuser and heat absorbing filter (Schott KG-1),
calculated from manufacturer data of sensor spectral response, transmission data of the Schott optical
correction filter, and measured transmission of PTFE diffusers.
Because of the limited view angle of the spectroscopy sensors (40°), it is crucial to add a light diffuser.
Two light diffusing materials were tested, PTFE and Opal cast acrylic sheet glass, and the transmission
measurements are shown in Fig. 9 (measured with the ASD FieldSpec).

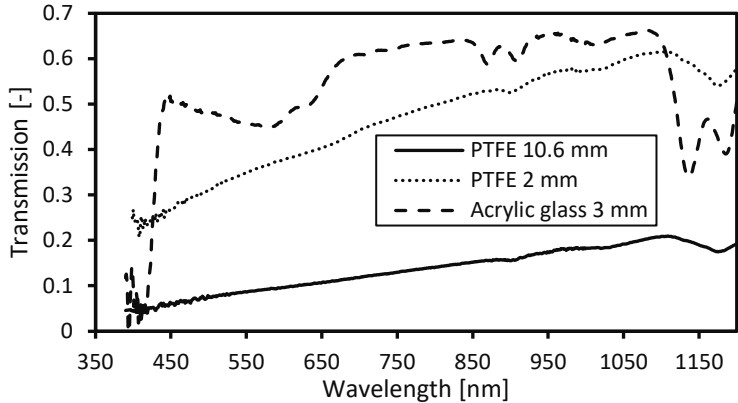


Figure 9: Transmission of PTFE diffusers and an Opal cast acrylic sheet glass diffuser measured with an
ASD FieldSpec spectroradiometer.
The reduced transmittance of the PTFE diffusers in the shorter wavelengths enhances the near infrared
crosstalk (see Fig. 6b top right panel and Fig. 8 top right panel). The combined effect of sensor and PTFE
spectral response with or without correction filters is shown in Fig. 10. Three versions of the
spectrophotometer were developed; one with a 10.6 mm PTFE diffuser to improve cosine response
(FROST1), a second version with a 2 mm PTFE diffuser and a correction filter on the blue sensor
(FROST2), and a third version with a 2 mm PTFE diffuser and correction filters on the blue and red
sensor (FROST3). The spectral selective quality on real world measurements (Fig. 10) was calculated
from the combined effect of the spectrophotometer filter characteristics (Fig. 6), diffuser (Fig. 9) and
Schott correction filters (Fig. 7).

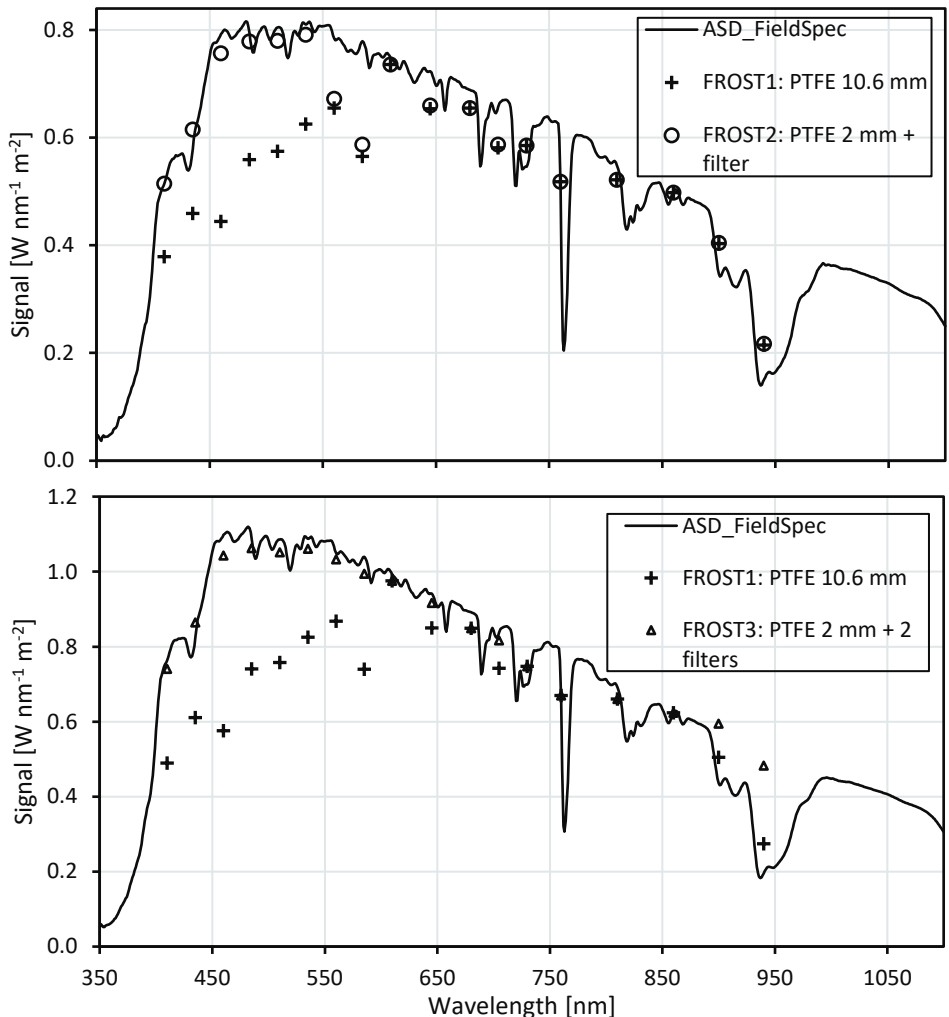


Figure 10: Outdoor measurements (ASD-FieldSpec) with calculated response of 3 FROST versions: FROST1 with a 10.6 mm diffuser, FROST2 with a 2 mm PTFE diffuser and a correction filter on the Blue sensor, and FROST3 with a 2 mm PTFE diffuser and a correction filter on the Blue and Red sensor, considering sensor spectral response and transmission of diffuser and correction filter, during clear sky conditions, Wageningen. Llower panel: 15 May 2022, 14:24 h UTC; upper panel: 11 March 2022, 13:35 h UTC.

Figure 10 shows that the first 6 channels, if uncorrected with a heat absorbing filter, underestimate the irradiance levels at the expected wavebands because these bands are very sensitive to the infrared region between 1000 and 1100 nm. At this infrared region, the solar radiation intensity is lower than what the blue sensor is supposed to see and thus leads to an underestimation of the Blue sensor for the visible channels. The heat-absorbing filters effectively remove this crosstalk. It also shows that the red sensor benefits from a heat absorbing filter for the wavebands 560 and 585 nm, but it greatly reduces the sensitivity of the 900 and 940 nm which makes the contribution of crosstalk from short wavebands too high (large positive deviation). Therefore the Red sensor should not be equipped with such a filter if the 900 and 940 nm wavebands are important, for example to estimate column atmospheric moisture (see Fig. 16).

The procedure to calibrate each waveband of FROST would require an accurate spectrophotometer and a clear day. First, each of the 18 PDF band responses of the sensor (with or without a correction filter!) and diffuser combination is multiplied with the known solar spectrum for a very clear day or measured with a calibrated spectrophotometer. This gives the nW m-2 reference value that the FROST sensor should produce for each waveband. Subsequently the FROST raw 18 wavebands outputs are multiplied

by the AMS calibration factors (since we use the uncalibrated output for fast measurement) and divided
by the reference values. The AMS spectroscopy sensor factory calibration values are written to the SD
card at the very start of the measurements. The derivation of the calibration values for each FROST
channel $i$ in [Counts W$^{-1}$ m$^2$] can be written as:
$$Cal_{FROST,i} = \frac{Counts_i \cdot Cal_{manufacturer,i}}{\sum_{\lambda_1}^{\lambda_2}\left[\frac{R_{sensor_{i,\lambda}} \cdot T_{diffuser_\lambda} \cdot T_{filter_\lambda}}{\sum_{\lambda_1}^{\lambda_2} R_{sensor_{i,\lambda}} \cdot T_{diffuser_\lambda} \cdot T_{filter_\lambda}} \cdot Source_\lambda\right]}$$
(1)

where $Counts_i$ is the signal output of a Frost channel $i$, from 1 to 18, [-], $Cal_{manufacturer\_i\_}$is the
manufacturer calibration factor [-] for channel $i$, $R_{sensor\_i,\lambda}$ is the normalized peak spectral response of
channel $i$ [-] at wavelength $\lambda$ [nm], $T_{diffuser,\lambda}$ is the spectral transmission of the diffuser [-] at wavelength
$\lambda$ [nm], $T_{filter,\lambda}$ is the  transmission of the (optional) crosstalk correction filter [-] at wavelength $\lambda$ [nm],
Source$_\lambda$ [W m$^{-2}$] is the output of the reference light source at wavelength $\lambda$ [nm] (preferably the sun), $\lambda_1$
and $\lambda_1$ are the lower and upper boundaries of the spectral sensitivity range (including crosstalk) of the
FROST. Note that the denominator is the spectrally-weighted source-signal strength.

The sensor output sensitivity is then expressed as counts per (W m$^{-2}$). The normalized sensor response is
provided in Fig. 6 and 8 and in Table s1 in Supplementary Materials. An example of sensitivity values is
presented in Table 2. These values were derived on 11 March 2022 13:35 UTC for the 10.6 mm diffuser
version and the 2 mm diffuser + 1 filter version. The 2 mm diffuser with 2 correction filters was
measured on 15 May 2022 (Fig. 10). Note that these values are only valid for an integration value of
13.9 ms and a gain of 16. We do not recommend to use channels with Flag 2 or 3 if spectral accuracy is
required.
Table 2: Sensitivity, or counts (*C*) per (W m$^{-2}$) of FROST1, -2 and -3 with different configurations, offsets
always zero. Every sensor uses a gain of 16 and an integration time of 13.9 ms. The flags denote quality
of measurement (waveband accuracy). Flag 0: low crosstalk, Flag 1: crosstalk<20%, Flag 2:
20%<crosstalk<35%, Flag 3: crosstalk>40%. The colors in the first column indicate each of the three
sensors in a FROST. The final column shows the improvement factor on sensitivity when using a 3 mm
white Acrylic glass diffuser instead of a 2 mm PTFE diffuser.

| Waveband [nm] | FROST1 PTFE 10.6 mm, Filter: no, Sensitivity [C W$^{-1}$ m$^2$] | Flag | FROST2 PTFE, 2 mm, Filter: on blue sensor, Sensitivity [C W$^{-1}$ m$^2$] | Flag | FROST3 PTFE, 2 mm, Filter: on blue and red sensor, Sensitivity [C W$^{-1}$ m$^2$] | Flag | FROST_AG Acrylic glass, 3 mm, Sensitivity increase $\dfrac{T_{Acrylic\ glass_{3mm}}}{T_{PTFE_{2mm}}}$ |
|---|---|---|---|---|---|---|---|
| 610 | 116 | 0 | 474 | 0 | 494 | 0 | 1.61 |
| 680 | 132 | 0 | 503 | 0 | 436 | 0 | 1.49 |
| 730 | 156 | 0 | 549 | 0 | 554 | 0 | 1.39 |
| 760 | 156 | 0 | 413 | 0 | 387 | 0 | 1.36 |
| 810 | 188 | 0 | 673 | 0 | 651 | 0 | 1.31 |
| 860 | 194 | 0 | 760 | 0 | 649 | 0 | 1.25 |
| 560 | 51 | 2 | 253 | 2 | 186 | 0 | 1.70 |
| 585 | 70 | 2 | 333 | 2 | 195 | 0 | 1.65 |
| 645 | 55 | 1 | 256 | 1 | 168 | 0 | 1.56 |
| 705 | 70 | 1 | 295 | 1 | 117 | 0 | 1.43 |
| 900 | 90 | 0 | 348 | 0 | 19 | 3 | 1.12 |
| 940 | 107 | 0 | 395 | 0 | 25 | 3 | 1.20 |
| 410 | 94 | 3 | 153 | 0 | 157 | 0 | 1.55 |
| 435 | 100 | 3 | 200 | 0 | 206 | 0 | 2.38 |
| 460 | 144 | 3 | 211 | 0 | 218 | 0 | 2.08 |
| 485 | 96 | 3 | 209 | 0 | 217 | 0 | 1.94 |
| 510 | 94 | 3 | 213 | 0 | 221 | 0 | 1.85 |
| 535 | 84 | 3 | 204 | 0 | 213 | 0 | 1.76 |

### 3.2 Temperature sensitivity and drift

The diffuser and sensor were both tested for temperature effects. The measurements were corrected for sensor temperature drift or drift in lamp output by measuring the lamp output with an extra sensor outside the oven. The oven has an internal fan to assure a homogeneous temperature within the oven chamber. The PTFE filter shows a significant jump in transmission around 21°C, then reaching a plateau and slowly declining past 35°C (Fig. 12, upper panel). The temperature was slowly increased and stabilized for 30 mins at each measurement point to minimize thermal delays in the PTFE material.

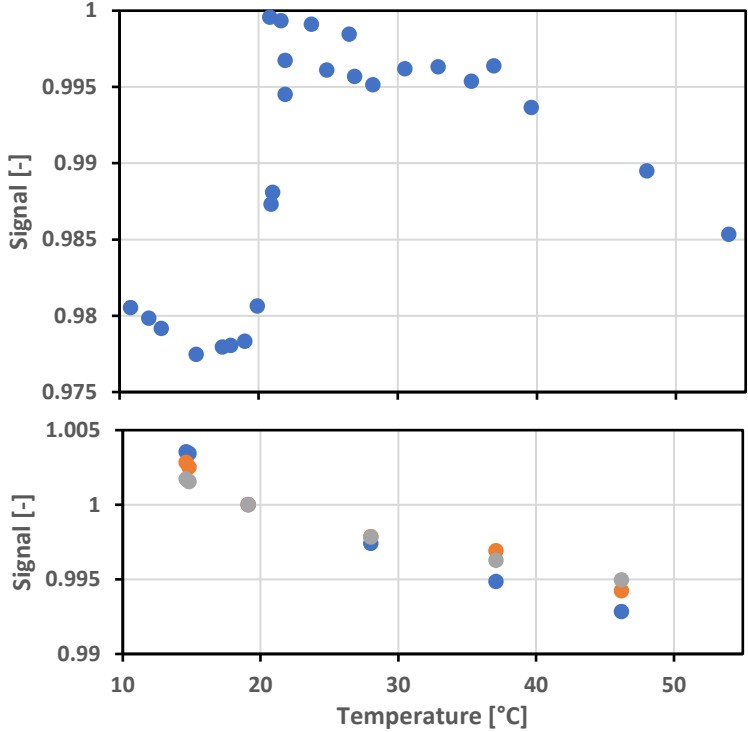

423

Figure 11: Temperature response; upper panel PTFE diffuser, lower panel 3 random light sensors (-250 ppm).

Three spectroscopy sensor chipsets (3x 18 waveband) were oven-tested for temperature sensitivity between 16 to 46°C. Overall temperature sensitivity is -250 ppm $K^{-1}$ with a small variation among the three sensors. Lower temperatures were not possible due to condensation issues when reaching the dewpoint temperature of the laboratory (Fig. 11).

### 3.3 Cosine response and GHI

The cosine response measurements (outside, LED lamp) had a better performance for the 10 mm diffuser but nevertheless had some inconsistencies among the three sensors. We tried to improve the cosine response by leaving part of the sides uncovered but this caused a very high asymmetry among the three sensors. The explanation is that the three sensors do not have the same viewing angle location under the diffuser, thus some will see more from the side than the other sensors. The side sensitivity is greatly reduced with a thinner filter but at the expense of a reduced cosine response (Fig. 12).

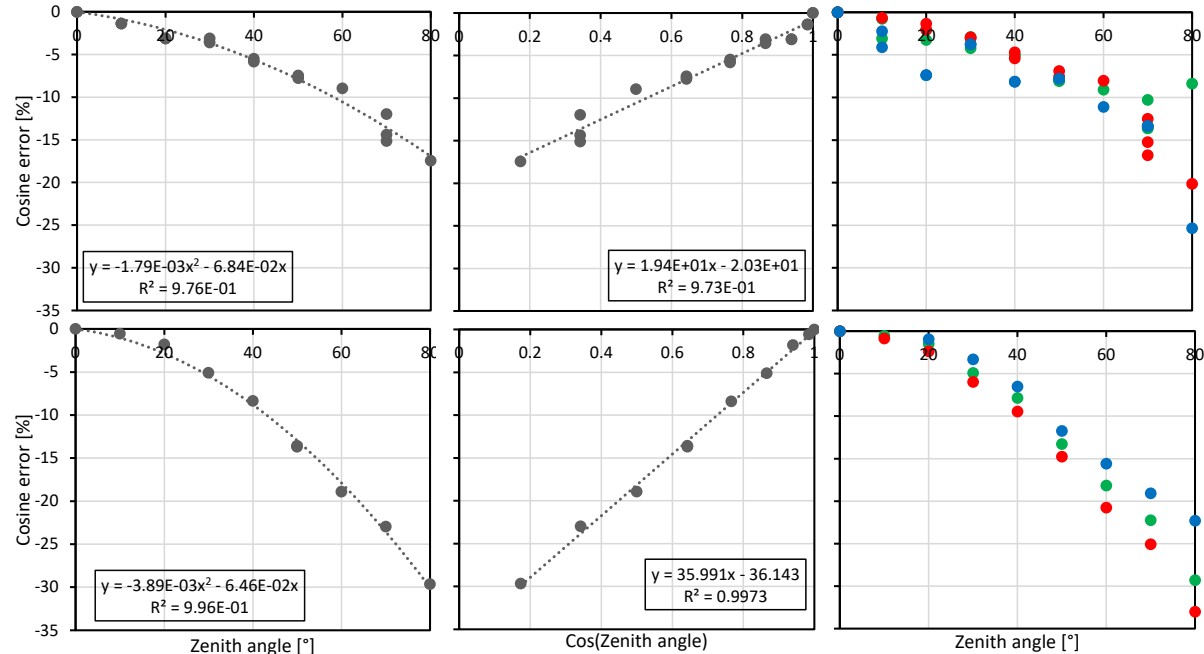

Figure 12: Upper graphs: 10.6 mm diffuser (black sides). Bottom graphs: 2 mm diffuser (sides painted black). The right panels are color-coded for each sensor integrated circuit.

We found that most of the cosine response errors can be corrected afterwards and is demonstrated for the 2 mm filter, which had the largest cosine response error but less transmission loss. The accurate measurement of GHI can be achieved by first correcting for the zenith angle response (see Fig. 13, middle lower panel) and subsequently applying a second order linear regression against a reference pyranometer on one clear day (19 March 2021). Additionally, a correction for the limited spectral response is needed. We tested this calibration method for the average signal of all 18 wavebands and on single wavebands. The dataset contains clear sky days (Fig. 13), overcast days (Fig. 14) and rainy weather (Fig. 15). The best overall results were achieved with either channel 645 or channel 705 nm, with residual errors mainly below 10 W m$^{-2}$ during contrasting weather conditions. Due to the spatial separation of 156 m between our sensors and the reference solar radiation measurements and the differences in response speed, we rejected the cloud passage time intervals. The 645 and 705 nm wavebands seems to correct cloud effects on the GHI where irradiance is enhanced below 500 nm and reduced due to water absorption bands at wavebands >1 μm.

The remaining uncertainty of the clear-day calibration (up to 10 W m$^{-2}$ or 5%)is mainly related to small levelling uncertainties or tolerances in input optics of both reference and our sensors. This is visible as a shift from a negative to a positive bias around 12 UTC (Fig. 13).

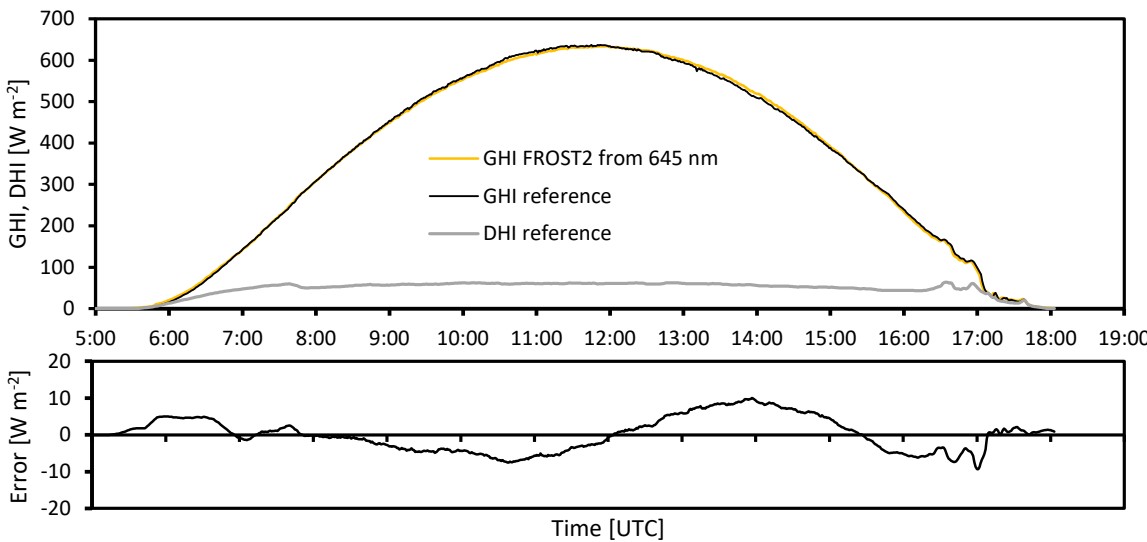


Figure 13: Comparison between GHI measured using a pyrheliometer and diffuse radiation sum (on a
suntracker and correcting for zenith angle)  and a calibrated FROST2 with 2 mm PTFE diffuser and 1
correction filter, diffuse horizontal irradiance (DHI) measured with pyranometer mounted on a suntracker
with shading ball, Veenkampen weather station, 19 March 2022. Relative errors at GHI >200 W m$^{-2}$ are
<2% and mainly related to horizontal misalignment causing an asymmetric error before/after noon.

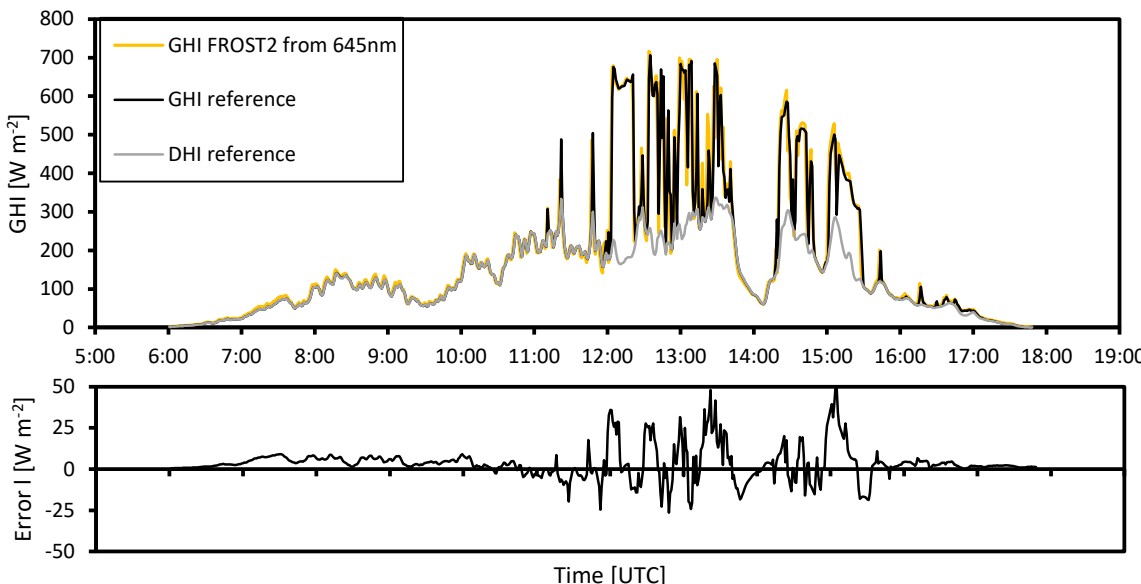


Figure 14: Calibrated FROST2 GHI from 645 nm (calibrated on a clear day, 19 March, see Fig. 13),
calibration tested with cloudy weather conditions. Cloudy weather conditions in the morning, some
clearing in the afternoon, 1 minute averaged data, error plot 10 min running mean to suppress
differences due to spatial separation of FROST and reference (145 m apart), Veenkampen weather
station, 14 March 2022. Relative errors at GHI >100 W m$^{-2}$ are <7% and mainly related to spatial
separation between FROST and reference.


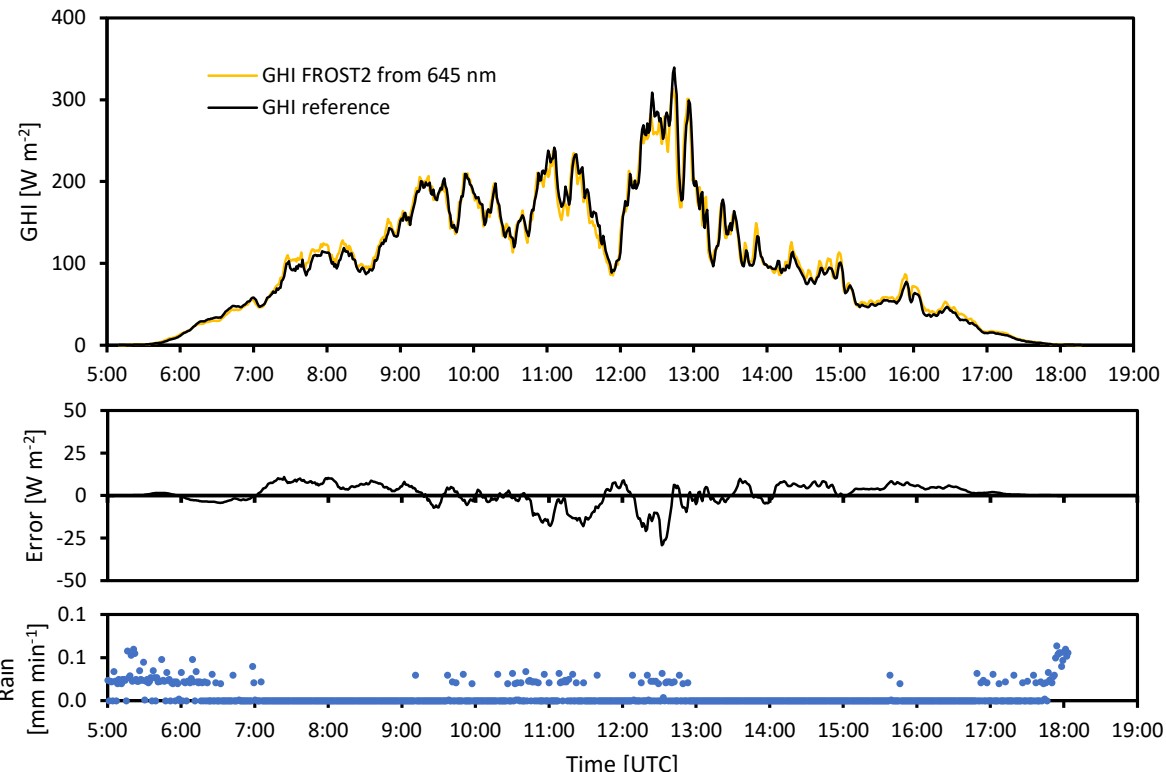


Figure 15: Calibrated FROST2 GHI from 645 nm (calibrated on a clear day, 19 March, see Fig. 13), calibration tested under rainy weather conditions, 1 minute averaged data, error plot 10 min running mean to suppress differences due to spatial separation of FROST and reference (145 m apart), Veenkampen weather station, 31 March 2022. Relative errors at GHI >100 W m$^{-2}$ are <7% and mainly related to spatial separation between FROST and reference.

The instruments were not dried during the precipitation event (Fig. 15 and 16). Water droplets on the diffuser may affect light transmission and diffuser optical properties. Note that in Figs. 13-15, the nocturnal offsets are zero.

Next, we investigated how the light spectra is modified by clouds or rain. The two instruments, one with a 10 mm diffuser (FROST1) and the second version with a 2 mm diffuser and crosstalk correction filter on the blue sensor (FROST2), were used to calculate the spectral change due to cloudy or rainy weather conditions (Eq. 2).

$$Spectral\_change_i = \frac{\frac{Counts_{i,clouds\_rain}}{\frac{1}{18}\sum_{i=1}^{18}Counts_{i,clouds\_rain}}}{\frac{Counts_{i,clear}}{\frac{1}{18}\sum_{i=1}^{18}Counts_{i,clear}}} \qquad (2)$$

Data from the Figs. 13-15 experiments were used and, of the three contrasting days, the 11-12 UTC intervals were averaged and normalized for the average spectral signal of the 18 wavebands. Figure 16 shows that the 940 nm waveband is very sensitive to moisture, with a reduction of more than 20% as compared to its nearest waveband. Accordingly, it can be used to derive information about atmospheric moisture such as column water vapor. Both cloudy and rainy conditions appear to modify the spectra in a similar way (Fig. 16). The low enhancement in the first four wavebands of the instrument with the 10 mm diffuser version (FROST1) is related to the strong crosstalk in the near infrared. The corrected version with the 2 mm diffuser (FROST2), which contains the crosstalk correction filter, shows an enhancement due to clouds and in line with the findings by Durand et al., 2021, who had an enhancement below 465 nm. The 645 or 705 nm as shown in Figs. 13-15 appear to have the right amount of sensitivity reduction due to clouds and rain (slightly stronger) to be used for GHI measurements. It is, however, recommended to use all 18 bands and use a proper weighting function

that reduces sensitivity in the visible region. We currently have no explanation for the enhancements
between 750 and 860 nm.

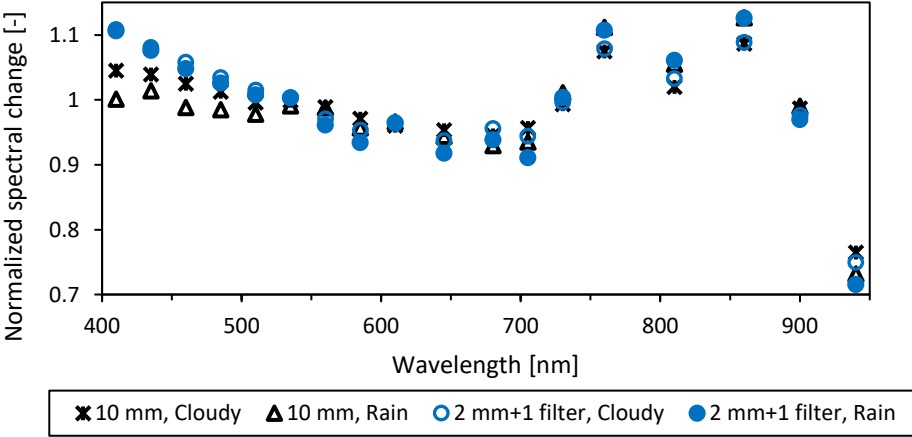


Figure 16: Two FROST instruments, one with a 10 mm diffuser (FROST1) and one with a 2 mm diffuser
and correction filter (FROST2). The normalized spectral cloud modification factor, is the spectral change
of cloudy (14 March, 2022) and rainy weather (31 March, 2022) compared to a cloud-free day (19 March
2022), Veenkampen weather station, data averaged between 11 and 12 UTC for each day.
The long-term drift was tested at the Lindenberg rooftop observatory. One instrument was measuring
from 22 June to 31 August 2021 (without any missing 0.1 s measurements). These 2.5 months of data
were converted to GHI values by using only one relatively clear day (13th August) and compared with
their reference pyranometer. The GHI standard error was 2.5 W m$^{-2}$ for daily averages with a diffuser
temperature correction obtained by increasing sensor values by 2% at temperatures below 21°C
according to Fig. 11 and cosine response correction according to Fig. 12 upper panel (for daily errors see
Fig. 17 upper panel). Additionally, the GHI deviations in percentage between 12 and 13 h UTC were
averaged to reveal possible sensor drift in time. The diffuser correction practically removed all long-term
drift (Fig. 17, middle panel).

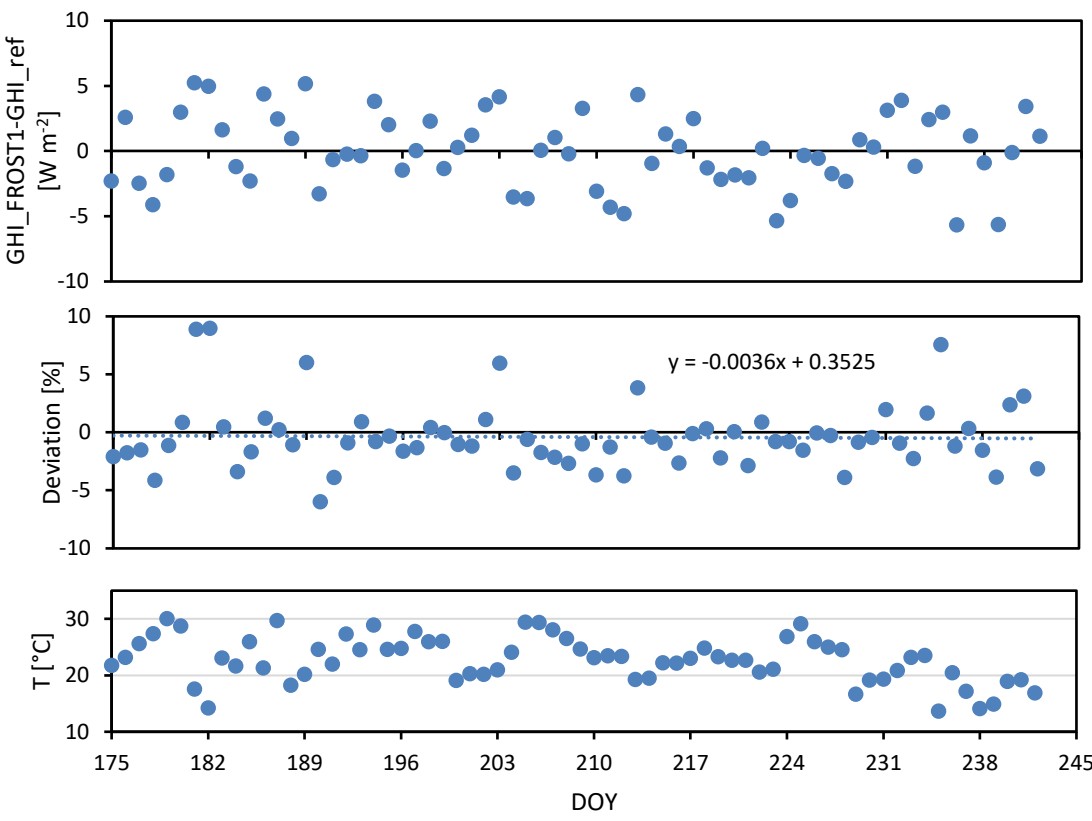


Figure 17: Long-term stability of FROST1 GHI measurements (using the 645 nm channel and calibrated
with DOY 226, 14 August data). Upper panel: daily average FROST GHI deviation from reference GHI.
Middle panel: FROST1 GHI deviation from averaged data between 12 and 13 h UTC. Lower panel:
average air temperature between 12 and 13 h UTC, during a 2.5 month comparison experiment at
Lindenberg. Measurements corrected for PTFE diffuser transmission change at 21°C.

**3.4 Spatial measurements and synchronization**
For spatial measurements, exact synchronization is essential. Our GNSS solution uses the hardware
timing pulse of the GNSS to trigger a measurement. To illustrate the synchronization performance we
set-up three stand-alone FROST sensors and let them run for 1 h outdoors. We then placed them in a
dark room and at 12:00:45.6 UTC a LED light source was switched on for 0.3 s. Figure 18 shows 1.1 s of
collected 10 Hz data of the 610 nm waveband. The response appears instantaneous and perfectly
synchronized. There is still an integration time for each measurement and this was set at 13.9 ms for
FROST s16 and s20 and, for testing purposes, twice as long for the experimental version with a less
transparent diffuser to get more signal. This instrument is denoted with "Exp" in Fig. 18. Therefore, the
"Exp" FROST occasionally showed a small delay and illustrates the importance of configuring all sensors
with the same integration time. Figure 18 also shows that the instruments have no zero offset (no dark
current) errors.

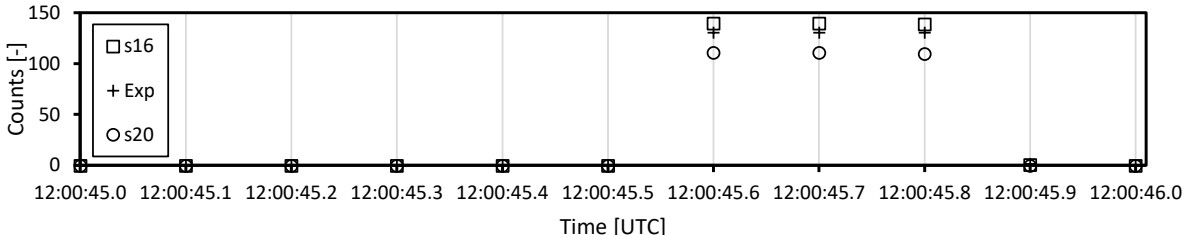

Figure 18: Example of synchronization, response speed and zero offsets of three standalone instruments (uncalibrated). All three use their own GNSS for synchronization. Light pulse of 0.3 s generated by a LED lamp.

The full sensor readout requires two integration cycles with each cycle measuring 12 channels (see Table 3). As a result, there is a maximum of one integration cycle delay between certain channels (with our default settings: maximum 28 ms). Six channels are measured twice within one default measurement cycle (Table 3). For critical synchronization applications, it is possible to measure only 12 of the 18 channels during each measurement cycle.

Table 3: Readout order during one full measurement cycle.

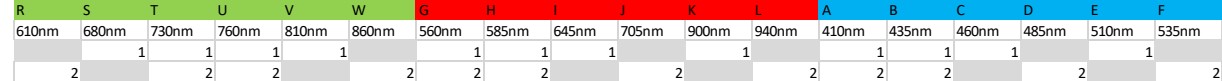

| R | S | T | U | V | W | G | H | I | J | K | L | A | B | C | D | E | F |
|---|---|---|---|---|---|---|---|---|---|---|---|---|---|---|---|---|---|
| 610nm | 680nm | 730nm | 760nm | 810nm | 860nm | 560nm | 585nm | 645nm | 705nm | 900nm | 940nm | 410nm | 435nm | 460nm | 485nm | 510nm | 535nm |
| | 1 | | 1 | 1 | 1 | | 1 | 1 | 1 | | 1 | | 1 | 1 | 1 | | 1 |
| 2 | | 2 | 2 | | 2 | 2 | 2 | | 2 | | 2 | 2 | 2 | | 2 | | 2 |

The downside of a fast integration cycle is a smaller output signal. The 10.6 mm diffuser reduces the light onto the detector significantly, approximately 120 to 30 counts per channel at 650 W m$^{-2}$. The 2 mm diffuser increases the signal by a factor of four. Longer integration times are considered but should be less than 50 ms to assure a sustained 10 Hz output (two integration cycles<100 ms). Additional time is needed for data communication. The AMS spectroscopy sensor output is in ASCII format and therefore more digits require more time to transmit.

For the measurement campaign in Falkenberg, a large 2D sensor grid was deployed with a 50 m grid spacing. It is a good illustration of the spatial dynamics of GHI during partly-cloudy conditions. The 1 min averaged data at one point shows the cloud enhancements and the 10 Hz measurements show the high dynamics and spatial variation along a 150 m transect (Fig. 19).

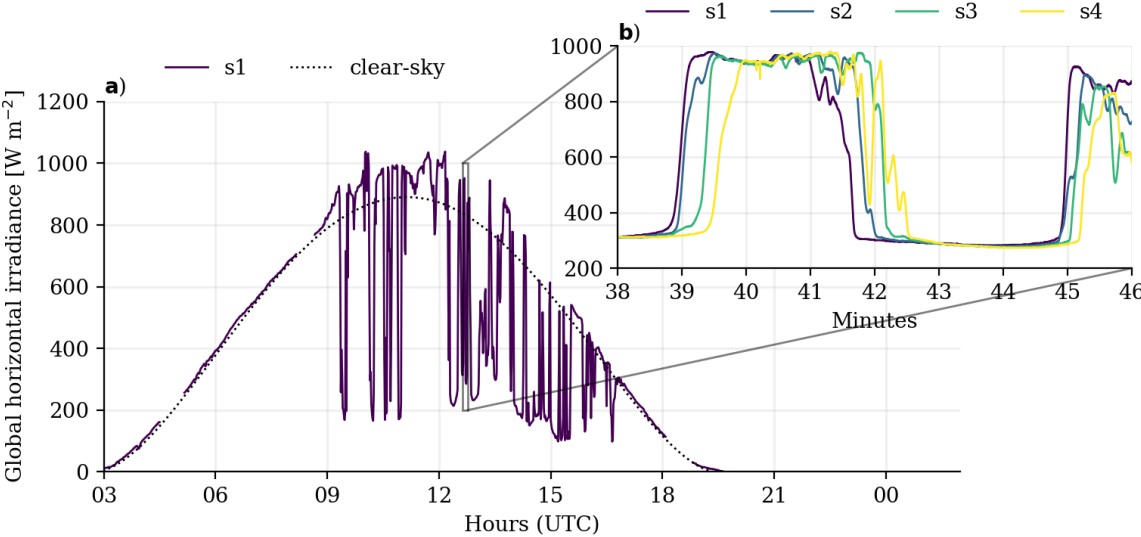

Figure 19: 10 Hz measurements of spatial variation of GHI at four locations along a 150 m west-east
transect (b) compared to one location (a) at 1 minute averages. The dashed line shows the CAMS
McClear clear-sky product. Falkenberg, 27 June, 2021.
**3.5 Photosynthetic Active Radiation**
Sensors for measuring Photosynthetic Active Radiation (PAR) are usually constructed using a silicon
photo diode and a light bandpass filter from 400 to 700 nm. Photosynthesis is a quantum process and
therefore measurement are usually expressed as a Photosynthetic Photon Flux Density (PPFD, μmol
photons $m^{-2}$ $s^{-1}$). The sensor therefore must account for the larger number of photons at larger
wavelengths. The wavelength sensitivity (per W $m^{-2}$ $nm^{-1}$) is such that the sensitivity at 700 nm
wavelength is 1.75 times larger than at 400 nm. In our case, we have 11 well-defined wavebands within
the PAR region. Therefore, a digital filter can be used to calculate PPFD. Since the sensor outputs in W $m^{-2}$ $nm^{-1}$, it must be converted to PPFD by calculating the number of moles per joule per waveband.
$^{2}$ $nm^{-1}$, it must be converted to PPFD by calculating the number of moles per joule per waveband.
The photon energy ($E_n$) at each waveband (n) is related to wavelength ($\lambda$) by the speed of light ($c$) and
the Planck constant ($h$): $E_n = \frac{hc}{\lambda_n}$. The photons ($P$) per m2 are: $P_n = \frac{R_n}{E_n}$, where $R_n$ is the irradiance measured
at each waveband. The number of moles is linked to the number of photons through Avogadro's number
($A$) and integration over all FROST3 11 wavebands yields the total PPFD:
$PPFD = (700 - 400) \int_{n=1}^{11} P_n / A n_{max}$ (2)

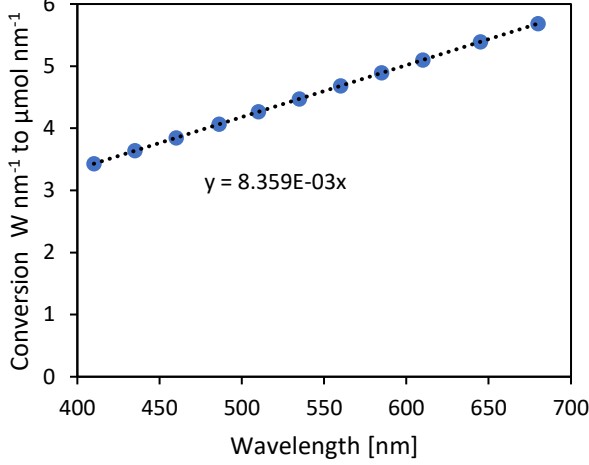


Figure 20: Conversion factor for each FROST3 waveband to calculate PPFD [μmol $m^{-2}$ $s^{-1}$].
The translation factor from W $m^{-2}$ $nm^{-1}$ to μmol $m^{-2}$ $s^{-1}$ $nm^{-1}$ for the 11 wavebands is depicted in Fig. 20.
The PPFD in Figure 10, lower panel, according to Eq. 2 is for example 1293 μmol $m^{-2}$ $s^{-1}$.
Note that the 11 channels within the PPFD range of FROST3 can be used to study vegetation specific
photosynthesis spectral response.
**3.6 Vegetation development**
The FROST was tested during a field experiment in La Cendroza, Spain (Lat: 41.692537, Long:
0.931540) (Liaise Campaign, Boone et al., 2021) from 14-22 July 2021. The instrument was placed on
the bare soil surface at the moment the Alfalfa vegetation started to develop. The Normalized Difference
Vegetation Index (NDVI) index (NIR-VIS)/(NIR+VIS) is used in remote sensing to quantify crop growth;
a low value is bare soil and a value of 1 represents a full-grown crop. It can be computed from at least
two wave bands, one in the near infrared (>700 nm) and a second waveband in the visible range. The
two wave bands 680 and 730 nm in version 1 are not affected by infrared crosstalk and both measure at
the same sensor chip to assure that both channels have the same viewing angle. Figure 21 shows daily
values of NDVI.

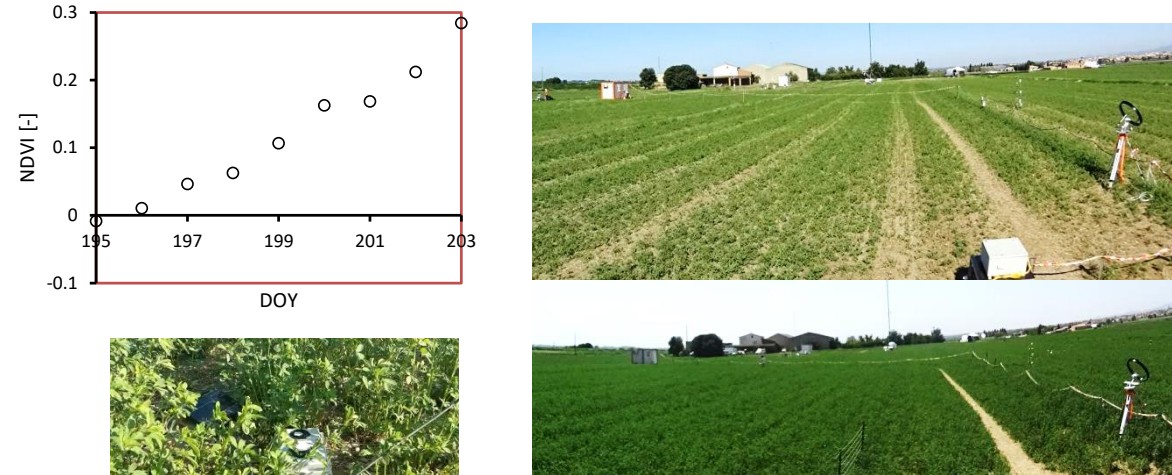

Figure 21: NDVI measurements calculated from FROST1 (with 10 mm diffuser) 680 and 730 nm
wavebands located inside an alfalfa canopy. Bottom left: FROST (top at about 8 cm height) placed on the
surface in between alfalfa, crop height 30 cm, (18 July). Top right: alfalfa crop on 14 July (crop height 23
cm). Bottom right: alfalfa crop (height 48 cm) on 22 July; Spain from 14-22 July 2022.
**3.7 Surface albedo**
A good test for the quality of the spectral measurements without having to deal with absolute calibration
uncertainties are surface reflectance measurements. The typical spectral reflectance signature of a
healthy vegetation has two minima in the visible range at 500 and 675 nm and a small peak at 550 nm.
Beyond 750 nm it is strongly reflective (about 50%). A bare soil surface, in this case a sandy soil patch
from a very deep soil layer that surfaced during the recent drilling of a well at the weather station,
served as a bare soil plot and had a negligible organic soil fraction. The ASD FieldSpec was equipped with
a cosine collector and operated in irradiance mode. Weather conditions were sunny with low soil moisture
content of the bare soil. The comparison is good considering the difficulty of sampling the same spot for
both instruments due to differences in cosine response, size of sensor head and levelling (Fig. 14).

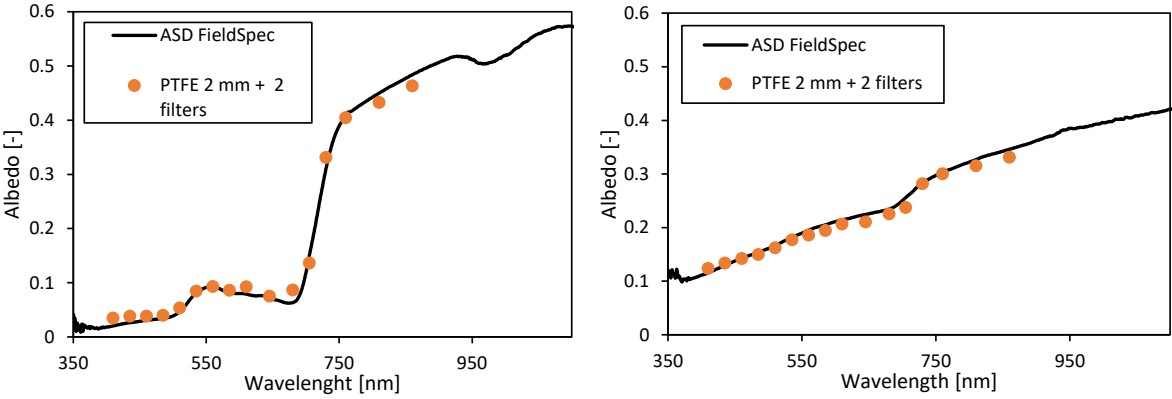


Figure 22: Spectral reflectance as measured by the FROST3 and the ASD FieldSpec with cosine collector.
Left panel: spectral reflectance of grassland, Veenkampen weather station, 14:12 UCT 15 May 2022,
Right panel: sandy soil (dry, no organic fraction), 14:06 UTC 15 May 2022.
The small underestimation of the 810 and 860 nm channels is related to the small cross correlation with
smaller wavelengths (Fig. 22).

**4. Concluding remarks**

The FROST instrument will enable new research opportunities. It is much faster than traditional
thermopile pyranometers and the low cost enables the deployment of large sensor grids. It can be
deployed very quickly because it is a fully stand-alone, "plug and play" solution and measurements are
always fully synchronized to UTC within at least a μs. The instrument has superior linearity (<0.2%), the
temperature coefficient is very low (-250 ppm $K^{-1}$), and was consistent among the three tested
instruments. In contrast to thermopile sensors, the FROST has no zero-offset errors. The drift with time
appeared insignificant during a 2.5 month field test. Compared to PAR sensors, FROST can resolve the
PAR spectra in 11 narrow wavebands (FWHM: 20 nm). This makes it possible to study wavelength
dependent photosynthesis responses of, for example, chlorophyl A and B. This is also relevant in canopy
profile studies where solar irradiance extinction through a canopy modifies its light spectra. The fast
response makes it possible to investigate the impact of the growth and wind induced movements of
vegetation on radiation fluctuations.
FROST measures GHI in 18 wavebands and includes a water absorption band, which makes it possible to
derive information about atmospheric moisture such as column water vapor. Additionally, by using
proper infrared crosstalk correction filters, it can monitor the spectral reflection properties of a surface
with its first 16 wavebands from 410 to 860 nm. FROST can also be used to monitor vegetation growth
by measuring NDVI.
We hope that other researchers will benefit from our crosstalk problem solution. Tran and Fukazawa
(2020), used the same AMS spectroscopy sensor to determine optical properties of fruit, but they did not
use LED light sources as recommended by the manufacturer. Their halogen light source emits much
infrared light >1000 nm and therefore the 6 channels of their blue sensor and 2 of their 6 red sensor
channels were greatly affected by infrared crosstalk. This is something they may not have been aware of
because the AMS spectroscopy sensor datasheet does not show the filter response above 1000 nm. We
believe their instrument performance would improve using our proposed correction filters.
Although the proposed cosine correction appears to give good results (2.5 W $m^{-2}$ standard error for daily
averaged GHI), we will continue to improve the cosine collector for large zenith angles. PTFE as a diffuser
material has better transmission properties below 400 nm than opal cast Acrylic sheet glass, but it
exhibits a 2% step wise increase in transmission beyond 21°C. Since the shortest waveband sensitivity
of the FROST sensor is limited to 400 nm (considering FWHM), we recommend using opal cast Acrylic
glass diffusers for future versions. This would also remove UV radiation exposure and reduce sensor
aging.

**Author contributions**

B.H. wrote the manuscript draft, methodology of synchronization, fast spatial spectral irradiance
measurements, instrument software, electronics and mechanical design, investigation and visualization
of spectral response, thermal sensitivity; W.M. organized the Germany and Spain field campaigns
including data organization and visualization of 2D performance in Fig. 19; W.M. and B.H. did the cosine
collector and long-term stability experiments and analysis; C.v.H. is the PI of the Shedding Light On
Cloud Shadows (SLOCS) project to which this research belongs and designed the research programme
that depends on this instrument; W.M. and C.v.H. reviewed and edited the manuscript.

**Acknowledgements**

We are grateful for the support and use of the optical calibration facility of DWD Lindenberg and many thanks go to Stefan Wacker and Steffen Gross. We also thank Harm Bartholomeus (Wageningen University, Remote sensing group) for providing the ASD FieldSpec and Emilie Wientjes (Wageningen University) for assistance with their Cary 4000 UV-Vis spectrophotometer. Simon Berkowicz is thanked for his valuable input and proofreading.

C.v.H., W.M., and B.H. acknowledge funding from the Dutch Research Council (NWO) (grant: VI.Vidi.192.068).

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
