# Peer review of "A new accurate low-cost instrument for fast synchronized spatial measurements of light spectra"

_EGUsphere, 2022_

## Author Response (AR1)

- **RC1**: 'Comment on egusphere-2022-726', Anonymous Referee #1, 05 Oct 2022 reply

*Our replies to the reviewer are presented below in italic font*

**General comments**

The article is correctly structrured. The scientific rationale is well defined as well as the methodological approach to answer to it.

The Discussion section should be more comprehensive; in particular it should include at least a qualitative assessement of the error budget components affecting the measured PAR with the FROST system.

*We agree that the Discussion section should be more comprehensive and will elaborate this section based on the specific comments from both reviewers. PAR measurements: The main error is the lack of spectral information between the 11 PAR wavebands. However, the information of 11 wavebands is also an advantage (compared to a single band) as it does provide spectral information of the PAR region.*

*See: track changes document: L639-684*

The interpretation of some results could greatly benefit with the inclusion of mathematical formulae. The radiometrically correct weighting of the wavebands spectral shape should be part of the FROST spectral calibration and digital filter calculations.

*The radiometrically correct weighting of the wavebands spectral shape was part of our calibration procedures; see L339-356.  We provided all waveband response data as digital data in supplementary material S1. We will include mathematical formulae.*

*See: track changes document L417 and L594.*

**Specific comments**

**Abstract**

- line 12: Doesn't the autor mean "... global horizontal irradiance (GHI)..." ?

*correct*

*See: track changes document: changed to GHI*

- line 19: is "... zero offsets ..." a synonim for dark current? If yes, please replace by it. It is the most used term in the radiometry field.

*Yes, we agree that the term dark current is actually describing well the mechanism that causes offsets in silicon light sensors, but we felt that "zero offsets" is more commonly used for GHI sensors. It can be changed into "dark current".*

*See: track changes document: added in line 554*

- line 23: 2% with respect to which reference temperature?

*It is indeed not clear if the jump happens in the cooling or heating direction around 21°C We will clarify this.*

*See: track changes document line 23-24*

**1 Introduction**

- line 55: specify that the temperature sensitivity of semiconductors is generally temperature dependent

*In line 55 we state: "..., and temperature sensitivity".*

- line 70: give value of relative (in percentage) rms error as well.

*See: track changes document: Michalsky et al, (1991) reference was missing and added in L737-738. They only provide rms errors.*

- lines 55 and 59: non-linear/nonlinear or non-flat?

*We agree that "non-flat" response would be more appropriate.*

*See: track changes document L56*

- lines 66 to 69: A figure with both response curves would be more suitable to understand the differences highlighted in this sentence.

*We prefer inserting a reference containing such a Figure. See Fig. 1 of Alados-Arboleda et al., 1995.*

*See: track changes document L70*

- lines 71 and 72: it is not clear why their similar performance is "surprisingly accurate". Please develop.

*This was developed in lines 73-79.*

**2.1 Light Sensor**

- line 132: Spectrometer is not the good definition for this instrument as it implies the existance of a scanning mechanism. AMS AS7265x would be better referred as a 18-channel filter radiometer or equivalent. Apply same correction throughtout the text.

*The more specific term for our sensor would be a "filter-based spectrometer", but it still qualifies as a spectrometer. We will clarify this in line 132. The filters are already described in line 136.*

*See: track changes document: L89: added "filter"*

- line 134: It says that there are 3 bands, RGB with 6 Si photodiodes with all the associated optics and electronics but then in line 146, it says that there is a challenge to couple everything to the same sensing area. This seems contradicting as there are 18 sensing areas (from line 134). Clarify this.

*No, there are no 3 bands, no RGB bands. The Red, Green and Blue are used to identify each of the 3 light detection chips. Each chip detects 6 light wavebands. The manufacturer (AMS) also identifies the 3 chips using the same color coding. We understand that this may sound confusing. Line 135 should clarify this, but based on the reviewer's comment, more info is needed.*

*See: track changes document: color coding improved, see Figs.*

- line 143: If it is a specification "+/-" shouldn't appear before 10nm FWHM, instead a tolerance, if existing, could be specified either in percentage or absolute.

*It was meant to indicate the width of each waveband (FWHM), therefore "...20 nm FWHM specifications" would be better.*

*See: track changes document L157 and added +/-10 nm center-wavelength specification*

- line 143: "AMS states that their filter stability (in time and against temperature) is not detectible but does not provide further specifications." While this is an honest statement from the author, this question might be of non-negligible importance: UV exposition is known to generally degrade materials over time with a possible impact on the measurement performance in this particular case. Mentioning this issue, even from a qualitiative perspective with bibliographical support, would be of great value to the article.

*Agree, we will add that the manufacturer warns against strong UV exposure as this could affect long term optical performance. The UV exposure is greatly reduced by the diffusers. We will add recommendations to select acrylic glass as a diffuser material for a new*

*FROST version. Another advantage of Acrylic glass is that is does not exhibit the transmission jump with rising temperatures at 21°C.*

*See: track changes document L669-671*

**2.1 Diffuser material**

- line 155: missing %; give more insight on the 1% to 4% transmission changes. Is this between the 3 diffusers, with/without spectral dependence?

*Yes, between the 3 PTFE diffusers tested by Yliantilla and Schreden, 2004. They did not find such a sharp increase in transmittance as we did. However, it is expected to be a sharp increase since the crystallization happens at one temperature. We therefore think that our experiment was more accurate because of the very slow increase in temperature during the experiment. They also found some spectral dependence, especially at temperatures above 29°C (0.7% point difference between 420 nm 580 nm from 29°C to 45°C). We will check our data to see how much this would affect our diffuser.*

*See: track changes document: L169-170*

- line 157: clarify what the author means by "without jump"

*Smooth linear dependency on temperature, without the sudden transmission jump as found in the PTFE diffusers.*

*See: track changes document L171-172*

**3.1 Spectral response and temperature sensitivity**

- general comment: the reader is not necessarily familiar with a Cary spectrophotometer. Please give some detail on the measurement setup (with references when possible) as the spectrophotometer is used for three different types of measurements: wavelength scale, filter transmission and cross-talk.

*We will improve the description of the Cary spectrometer setup. It was not used for cross-talk measurements. At the DWD the Cary was capable of producing the relevant wavelengths, but it was impossible to place the FROST inside the Cary light excitation chamber. They also did not manage to measure the transmission of the PFTE diffuser inside the chamber, but only the transmission of the crosstalk test filter could be tested (Fig. 4).*

*The cross-talk was determined by placing an optical LP filter (Fig. 4) in front of the FROST. Any signal that would be measured by the FROST was interpretated as near infrared cross-talk (see L257-271).*

*At Wageningen, the Cary photo spectrometer was limited to about 800 nm, and thus it was only used to test center waveband positions of the first 12 channels. We will specify this in the text. We therefore could not test near infrared cross-talk. In this case the Cary was used as a monochromatic light source only.*

*See: track changes document: technical specs updated, Fig. 5 caption expanded, added L262-266*

- general comment: was the FWHM also measured with the Cary spectrophotometer or otherwise?

*Yes, but only for a few channels. These measurements confirmed for these channels the correct FWHM and therefore we trusted the manufacturer-supplied response curves (Fig. 6, left panels). The test with the optical LP filter (Fig. 5) confirmed that the near infrared crosstalk was not understated by the manufacturer (these response curves were provided upon personal request, it would have been nicer to find them in the manufacturer published datasheets).*

*See: track changes document: Fig. 6a added*

- lines 247 and 248: Does this wavelength accuracy refer to the position of the bandpass central wavelengths? If yes the measured wavelength could be shown, for instance in table 2, against the nominal wavelength.

*Yes, this is where we will explain that not all channels were tested, see previous comment.*

*See: track changes document: Fig. 6a added*

- lin 249: Sentence starting by "Comparison" is inconsequent. Rephrase it, putting it in relation with the follow-up sentences.

*Yes, changed to: Linearity was tested by comparing the FROST measurements against a reference thermopile pyranometer CM21 (Kipp and Zoonen, The Netherlands) and a stabilized halogen light source.*

*See: track changes document: L273-276*

- line 252 and 253: It would be intersting to have a brief summary of the details on how this non-linearity was obtained.

*We will provide an xy Figure to show the quality of the comparison (supplementary materials)?*

*Unfortunately the experimental data could not be retrieved*

- line 253: to which quantity does the calibration refer?

*Good point, it refers the light intensity response.*

*See: track changes document: L279-280*

- line 272: the paragraph starting here should be rearranged: first mention how the data in Fig.5 were obtained then proceed to compare and then to try to explain the differences observed.

*Lines 257-259 seem too limited to explain how the data was obtained. It was a test to find the intensity of cross-talk resulting from a FROST excited by a light source that only emits light >1000 nm.*

*See: track changes document: some rearrangements and additions: L284-307*

- line 272: if you know the Sun spectrum and the Xe lamp spectrum would not it be possible to correct for this?

*Yes, certainly. For clear sky conditions, measuring incoming solar radiation could work. But as soon as clouds appear it would deviate.  Additionally, it does not work for the applications mentioned in Chapter 3 such as measuring reflected solar radiation or surface albedo measurements, or NDVI measurements or PAR within a canopy (unknown spectral properties of leaves, surfaces).*

- line 306: why would the below crosstalk be increased? Additionally it could be intersting to point the origins of the below crosstalk earlier in the text.

*Agreed, additional explanation was lacking. We need to correct a mistake in the Figure reference;  6 should be 8.*

*See: track changes document: improved Figs. 5, 6b, changed: L348-352*

*Some further explanation: See Figure 6 lower right panel, cross-talk below waveband is significant. A correction filter would also greatly affect the response of the 900 and 940 nm bands and therefore increases the cross-talk of the lower wavelengths. This can be seen very clearly in Fig. 10 lower panel (circles).*

- figure 8: it would be more consistent to mantain the strucutre of figure 6: spectral responses in left column and crosstalk on right column

*Yes, we agree*

*See: track changes document improved Fig. 8.*

- figure 8: from the colors of crosstalk the reader might understand that the bars represent only "above" crosstalk. While this is probably the case, should be indicated in the legend.

*Yes, and "below" crosstalk is a very low fraction of total crosstalk. We will specify this.*

*See: track changes document: improved Fig. 8*

- line 318 and 322: Repeated phrase. Keep one instance and develop the sentence.

*Yes, the sentence "Note that it enhances infrared crosstalk can be omitted".*

*See: track changes document: Line removed and L385 Figure references added.*

- line 342: I understand the principle procedure of calibrating the FROST sensors with the solar spectrum, however some details are missing. For instance the weighing of the solar spectrum by the respective channel response function is not mentioned. This particular paragraph should include an equation detailing the calibration procedure.

*Details regarding the calibration method considering the weighting of the solar spectrum by each respective channel are not missing: Line 340 explains that the reference spectra (either from the sun or a reference spectrophotometer) should be multiplied by the PDF band responses of the FROST. We will add a reference to the Supplementary data (S1) of the PDF of the FROST response. Unfortunately, S1 only contains the PDF for the sensor only, without the transmission data of the correction filter or diffuser. This will be added to S1.*

*See: track changes document: L418-436 and new S1.*

- Table 2: Counts is usually untiless, but it is indicated has having nW$^{-1}$ Please clarify the quantity.

*These are calibration constants, and the units are [Counts nW-1], dividing the sensor signal by these values gives the output in nW for each waveband (Counts are not in nW-1).*

*See: track changes document: improved Table 2*

- Table 2 legend: How is the 35%<crosstalk<40% category handled?

*If waveband accuracy is important, reject the channels with Flags>0.*

**3.3 Cosine response and GHI**

- general comment: As this is used often during this paragraph, how to decouple GHI and diffuse radiation?

*The FROST cannot decouple diffuse radiation from GHI.*

- 13, 14 and 15: It is always more tangible if the error is expressed in relative units. Please show the the error in percentage. Additionally mention the content of each panel.

*Agreed, we will add relative units also and improve Figure 13, 14 and 15 captions.*

*See: track changes document: L481*

- 13, 14 and 15: The caption and the legend do not agree. What are each of the 3 curves in the top panel? What error is shown in the bottom panel?

*We will expand the Figure captions (for example diffuse radiation (Qd) in Figure caption is missing). Figures 14 and 15 captions will also contain the information provided in Figure caption 13.*

*See: track changes document: All Figs. 13-15 and Fig. captions improved*

- 15: Why are there only binary rain values, 0.00 or ~0.25 mm/min? The visualization of the bottom panel is confuse, please simplify.

*The rain gauge produces a much better resolution than 0.2 mm and outputs the data each minute (Ott Pluvio2s). We will correct this error and add rain gauge specifications and update the Figure.*

*See: track changes document: improved Fig. 15*

- line 407: Do you refer to figure 15 or 16? Does this information provide an additional insight on the data interpretation? If yes, please state which.

*We will change the reference; it should refer to both Figures 15 and 16. We will discuss the possible impact of water droplets on light transmission of the diffuser.*

*See: track changes document: L505-506*

- line 408: The paragraph starting here needs a mathematical expression to better support the text and the interpretation of Fig. 16.

*To improve Fig. 16, we will change the y axis legend. It should read "Normalized spectral cloud modification factor", see Durand et al., 2021.*

*See: track changes document: L507 added and Figure caption improved*

**3.4 Spatial measurements and synchronization**

- 18: There are only nine data points in this plot. Why not a table or a plot with improved readbility? The offset effect of the zeros values worsens the understanding.

*This Figure is a true time series of raw data output of 3 FROSTS responding to a step change in light intensity. There are no offset effects we can think of. The caption may need to reflect this in a better way, so we will add that the first 6 and last 2 samples are zero (no offsets). Of course, we could provide a table of these data also but it would just contain zeros at times 12:00:45.0, 12:00:45.1, 12:00:45.2, 12:00:45.3, 12:00:45.4, 12:00:45.5, 12:00:45.9, 12:00:46.0.*

*We think it is nice to show this as a Figure since it directly visualizes the perfect synchronization, the fast response speed and no zero offset (or dark current).*

**3.5 Photosynthetic Active Radiation**

- 1st paragraph: there seems to be a confusion between solar spectrum ($W.m^2.nm^1$), the number of photons of this spectrum per wavelength and the detector sensitivity. Please clarify the paragraph and taking into account the following remarks:
    - line 480: **measurements** refer to what?

*Measurements of PPFD intensity*

- lines 481 and 482: the number of photons at a given wavelength is unitless, while $W.m^2.nW^{-1}$ are the usual units of spectral solar irradiance.

*Removing (per W m-2 nm-1) solves this confusion.*

- lines 482: the wavelength sensitivity or the spectral dependence of the solar spectrum/number of photons?

*The number of photons as a function of wavelength (at constant W m-2 nm-1).*

- line 486: wavelength ($\lambda_n$)

*Accepted*

- second paragraph: the expressions and consequent calculations seem to not take into account the non-finite character of each of the FROST channels spectral reponse. All these quantities must be properly weighted by each of the wavebands spectral response. This paragraph should be rewritten taking into account the above commentaries and be accompanied by a more rigorous mathematical formalism and notation.

*No, we do consider the spectral weighting (assuming a calibrated FROST, see calibration L339-356). What cannot be considered is the lack of spectral information in between the spectral bands.*

*See: track changes document: New Eq. 1*

- 20: Offset term should be shown and, if zero, mentioned.

*Accepted*

*See: track changes document: Improved Table 2*

**4 Discussion**

- An overview of the factors contributing to the PAR measurement uncertainty should be expanded.

*Agree, the major factor would be the limited coverage of the PAR band due to narrow band response of the 11 bands.*

**Technical corrections**

- line 14: delete "very"

*The FROST covers the whole PAR range, but not the whole solar spectrum. We think it is unique for a low-cost PAR sensor to be able to quantify the light spectra within the PAR range for 11 wave bands.*

- line 69: their main uncertainty **is** related

*Accepted*

- Numbering of subsections: 2.1 is repeated

*2.1 should be 2.2 and 2.2 should be 2.3 (2.3 was missing)*

- line 100: verify autor name. Probably Lopes Pereira.

*Accepted*

- line 180: 5 m

*Accepted*

- line 252: a t => at

*Accepted*

- line 238: missing )

*Accepted*

- line 341: spectra is the plural form of spectrum.

*Accepted*

- line 352: check English

*Accepted, "recommend to" was missing*

- figure 10: correct formatting of units: nm-1 => $nm^{-1}$ for example. Correct the many instances thoughout the text.

*Accepted, subscripts somehow got lost during the submission process.*

- line 595: double check authors surnames: probably Peireira => Pereira, Goncalves => Gonçalves, Vazao => Vazão

*Accepted*

- sections 3.1 and 3.2: Temperature sensitivity is repeated

*Accepted: "3.1 Spectral response"*

***Our replies to the reviewer are presented below in italic font***

The manuscript describes the design and performance for a new solar radiation sensor ("FROST") with spectral and broadband/hemispheric sensitivity. While there are limitations to the sensor, it is very low-cost and also provides some technical advantages in response time that the design team exploits. I agree with the design team that FROST could be of wide interest for several applications. The manuscript is suitable for publication in AMT but requires revision largely to increase clarity and provide missing information. The text is straight-forward, but could use improvements in organization. As I read the manuscript, I made many comments and notes only to find the answer appear much later. For example, there are two separate sections describing the cosine response. Consolidation of like sections and discussion would help. Additionally, I would like to see more of the analysis defended with quantitative data: For example, what were the results of the cosine tests in Section 2.4? My specific comments are as follows:

*Based upon the valuable feedback received from the two reviewers, we agree the manuscript organization can be improved. We think that by addressing comments related to confusing sentences, sections, or figure references, we can improve the overall structure and clarity of the manuscript. This includes elaboration on the Discussion section as suggested by Reviewer 1, and e.g. moving certain paragraphs to a different section as suggested by Reviewer 2 below.*

*Regarding the cosine response tests in Section 2.3, the article is organized such that Section 2 describes the instrument design and measurement method. Results can be found in Section 3 (see Section 3.3 and Fig. 12.*

The manuscript requires copy editing: There numerous grammatical errors, unnecessary words, errant spaces, spurious pluralization, awkward wording, etc.

*Agreed, we will proofread the manuscript by a native speaker*

L59: I don't understand what you mean by "their vision".

*It refers to Martinez et al. 2009 (the second "their version" will be removed).*

L73: I know it's defined in the abstract, but you should also define GHI here.

*Accepted*

*See track changes document line 60*

L93-100: Doesn't this paragraph belong in Section 2?

*Good suggestion, we will work on this to further improve the manuscript organization.*

*See track changes, moved to section 2.*

Section 2: In Section 2 there are a lot of unanswered questions. For example, What serial protocol(s) the system uses for external communication? How data is archived; format, volume etc. What are the temperature limits on components/power? What is the transmittance of the diffuser? How long does the battery last? Many of these questions are answered later in Section 3. Either general reorganization to Sections 2 & 3 or some additional text in Section 2 describing where more details will be found later is needed.

*These issues seem related to the organization of the manuscript so we will improve it. We explained the Serial protocol in L197-198.*

*Data archival was explained in Section 2.5*

*Diffuser transmittance was explained in Fig. 9.*

*The battery run time can be inferred from the information in L121-123. On battery power supply, it would last about 40 hours. This will be added to the text.*

*See track changes document line 125.*

What are the temperature limits of components/power?

*We did not test temperatures below 5°C, but we expect that low temperatures will impact the LiPo battery capacity.*

*See: track changes document: L148 sensor only has an operational range from -40°C to 85°C*

*Experimental results are presented in Section 3. We think it fits better in Section 3 than in Section 2. But we will add additional text in Section 2 that diffuser transmission data can be found in Section 3.*

Section 2: Check the numbering of your subsections.

*Yes, second 2.1 should be 2.2 and 2.2 should be 2.3*

L179-186: What were the results of LED test? Can you provide a figure and analysis?

*See Section 3 with results of experiments in Fig. 12.*

Lines201-210: The commercial grade SDcard seems like a significant and critical vulnerability given these fast read/writes. Have you considered more robust (and more expensive) aMLC/SLC industrial versions that might be more reliable?

*We did not consider other SD cards since we never encountered a missing line of data, not even during the 2.5 months of continuous measurements, see L429. We think this shows the reliability of the commercial grade SDcard. Note that we are dealing with very slow write times. We are writing less than 2 KB per second whereas our SDcard can handle up to 85 MB per second. The large capacity of 32 GB means that this card will not wear down fast. We think it is useful to add this information to the manuscript.*

*See: track changes document: L221-223*

L214: This DOI is password-protected. I can't get the Restricted Access for Review link to work. It's unclear to me if this information is to be open access or not.

*We explicitly added L215-219 to allow open access for anyone that reads the manuscript. After final acceptance, the reference in L213-214 will be open access (it is the same Zenodo record).*

L235: The BSRN reference is Driemel et al.: https://essd.copernicus.org/articles/10/1491/2018/

*Accepted, we will add this.*

L242-243: Are these filters the ones that were first discussed at L169-177? I think a little more clarity is needed here.

*We will clarify this in more detail. The filter in L169-177 is only used to test crosstalk since it blocks all light below 1000 nm.*

L250-254: Can we see these results in a figure?

*Yes, we will provide this.*

*Experimental data could not be retrieved.*

Figure 5. What are the units? This is a fraction? Can you be explicit in the caption? Maybe this is defined later at L295 and that definition could be moved up to Figure 5? That said, I don't understand the definition. Further, because the radiance at the observed and interfering bands are different, I'm unclear how the fraction translates to a radiance bias.

*Figure 5 shows a fraction [-], unitless. The bias is explained in L270-280. We will provide some additional text because we realize it can be confusing. At 410 nm the crosstalk is about 55%, it means that when the sensor is illuminated with the xenon light source 55%*

*of the sensor signal is from the near infrared (>1000 nm) and not from the 410 nm waveband. We can add some text in the figure caption to explain this further.*

*We realize that the crosstalk definition is different for Figs. 6 and 8, since they show crosstalk for a flat spectrum, so will clarify this.*

*See: track changes document: L395-410, improved Fig. 8 and Fig. 6b, L305-307*

L273: Wouldn't cloudy conditions actually increase crosstalk because the incident IR increases at all wavelengths?

*Clouds do increase thermal infrared but reduce the near infrared waveband that are responsible for the crosstalk (see L392 or Durand et al., 2021).*

Figure 6. For clarity, please label the green, blue, red sensors on the figure since this is how you refer to them in the text

*This is a very good suggestion!*

*See: track changes document: Figures updated with color labelling.*

Figure 8. I would like to know the performance cost incurred for including the KG filters. Two comments. First, please show the results for the KG1/red. This is particularly important because of the substantial transmission losses in the red band. Second, I'm also concerned that the results as depicted are deceptive. The responsivity is shown in normalized rather than in absolute units. Therefore, with the filter applied and the crosstalk removed, the in-band responsivities appear to increase, but they should decrease and I would like to know by how much.

*The KG filters do reduce the in-band signal levels, as can be seen in the KG transmission curves in Fig. 7. Considering the red sensor with a KG-1 filter: 940 and 900 nm are reduced such that the crosstalk from shorter wavelengths become very dominant, see L336 and Fig. 6 lower right panel. The 560 nm and 585 nm wavebands lose about 5%, the 645 nm about 20% and the 705 nm about 40% signal but it greatly reduces crosstalk.*

*We will change Fig. 8 into the same formatting as Fig. 6 and add the red sensor including a KG-1 filter.*

*See: track changes document: improved  Fig. 8 (now with red sensor included), improved Table 2, shows impact of transmission loss.*

L318-319: Why does the diffuser add crosstalk? Is it because the more light is collected at large zenith angles where the infrared signal is larger? Is it then primarily a clear-sky problem?

*No, see Fig. 9, the diffuser absorbs a lot more in the shorter wavelengths and therefore enhances the impact of infrared crosstalk (see also L322).*

L325: Since you are comparing three versions, I feel like there should be three sets of symbols in Figure 10 comparing to the field spec, but I only see 2 (circles and pluses corresponding to versions 1 and 3).

*Good point, we will change the circles in the lower panel (it is the third sensor version).*

*See: track changes document: Figs. Improved and FROST serial numbers introduced (1,2,3).*

L327: I'm confused about the world "calculated". Figure 10 are all measurements, yes?

*No, the FROST response is calculated from the measured spectrum multiplied by the normalized response curve and considering transmission of diffuser and KG filters. For example, the underestimation of the blue sensor without correction filter and with a thick PTFE diffuser (as in Fig. 8 left panel) would be more sensitive for the NIR for most of its channels and therefore it underestimates the solar radiation at the expected wavebands. Figure 19 clearly shows a real-world example of how much the waveband improvement is using the KG filters.*

Figure 13, 14, 15: Can you be more explicit in the legend about which sensor is which?

*Yes, this is clearly missing.*

*See: track changes document: Figs. Improved including captions.*

Sections 4 & 5: I think these can be just one section.

*Agreed*

Section 3. The stated purpose of the diffuser is to increase the sensitivity of the sensor from the nominal 41 deg FOV to the hemisphere but it seems it should also provide the advantage of collapsing the cosine response function to a constant, specifically its value at 45 deg, the effective diffuse angle (Vignola et al. p.158). Notably, this effective angle is still outside the FOV and the consequences of that are not obvious to me. Can you comment on this?

Vignola, Frank, Joseph Michalsky, and Thomas Stoffel. Solar and infrared radiation measurements. CRC press, 2019

*We do not have access to the book you refer to. The diffuser increases the field of view to 180°; ideally the sensor should respond have a cosine response curve. For example. light at the 45° zenith angle should output 70.71% of the signal it would receive if the light source would be placed at the zenith angle).*

---

## Referee Report (RR1)

Abstract

1) See: track changes document: L639-684

   OK

2) See: track changes document L417 and L594

   I have some doubts concerning equation 1 (L417)

-   How can a calibration factor be unitless? Or is it just a multiplicative gain-type factor?

-   If Rsensor is the spectral response, it also should have absolute units, otherwise it is just a *relative* spectral response.

-   On the form of the equation: if the wavelength dependence of *Rsensor* is expressed - Rsensor_i,$\lambda$ – so should be the case for *Tdiffuser* and *Tfilter*.

3) See: track changes document: changed to GHI

   OK

4) See: track changes document: added in line 554

   The remark refers to line 19. There is no mention of dark current nor zero offsets at line 554. Please verify back line 19.

5) See: track changes document line 23-24

   OK

1. Introduction

6) In line 55 we state: "..., and temperature sensitivity".

   OK

7) See: track changes document: Michalsky et al, (1991) reference was missing and added in L737-738. They only provide rms errors.

   OK

8) See: track changes document L56

   OK

9) We prefer inserting a reference containing such a Figure. See Fig. 1 of Alados-Arboleda et al., 1995.

OK

10) This was developed in lines 73-79

OK

**2.1 Light Sensor**

11) The more specific term for our sensor would be a "filter-based spectrometer", but it still qualifies as a spectrometer. We will clarify this in line 132. The filters are already described in line 136. See: track changes document: L89: added "filter"

In order to qualify as spectrometer, an instrument should measure a wavelength dependent quantity (units ~ W.m-2.nm-1). FROST is measuring integrated signal in 18 different wavebands (units ~ W.m-2). Please note that this is in no way diminishing FROST instrument general quality. It is just a matter of correctness of radiometric definitios.

I advise to revise several instances of document in the sense of denominating FROST as a multi-channel radiometer rather than a spectrometer.

12) No, there are no 3 bands, no RGB bands. The Red, Green and Blue are used to identify each of the 3 light detection chips. Each chip detects 6 light wavebands. The manufacturer (AMS) also identifies the 3 chips using the same color coding. We understand that this may sound confusing. Line 135 should clarify this, but based on the reviewer's comment, more info is needed.

See: track changes document: color coding improved, see Figs

I don't understand if this was corrected as there is no information concerning this on line 135.

13) See: track changes document L157 and added +/-10 nm center-wavelength specification

OK

14) See: track changes document L669-671

OK

**2.1 Diffuser material**

15) See: track changes document: L169-170

OK

16) See: track changes document L171-172

OK

**3.1 Spectral response and temperature sensitivity**

17) See: track changes document: technical specs updated, Fig. 5 caption expanded,

added L262-266

OK

18) See: track changes document: Fig. 6a added

OK

19) See: track changes document: L273-276

OK

20) We will provide an xy Figure to show the quality of the comparison (supplementary

materials)? Unfortunately the experimental data could not be retrieved

OK, but the method should be briefly explained nonetheless

21) See: track changes document: L279-280

OK

22) See: track changes document: L279-280

OK

23) See: track changes document: some rearrangements and additions: L284-307

OK

24) See: track changes document: improved Figs. 5, 6b, changed: L348-352

OK

25) See: track changes document improved Fig. 8

OK

26) See: track changes document: Line removed and L385 Figure references added

OK

27) See: track changes document: L418-436 and new S1.

OK

28) See: track changes document: improved Table 2

OK

**3.3 Cosine response and GHI**

29) The FROST cannot decouple diffuse radiation from GHI.

OK

30) Agreed, we will add relative units also and improve Figure 13, 14 and 15 captions

The error is still shown in absolute units [W.m-2]. I think it would be more readable if it is given in percentage.

31) See: track changes document: All Figs. 13-15 and Fig. captions improved

OK

32) See: track changes document: improved Fig. 15

OK

33) See: track changes document: L505-506

OK

34) See: track changes document: L507 added and Figure caption improved

OK for changes in Figure 16. But I would still strongly recommend including a mathematical expression.

**3.4 Spatial measurements and synchronization**

35) We think it is nice to show this as a Figure since it directly visualizes the perfect synchronization, the fast response speed and no zero offset (or dark current)

OK

**3.5 Photosynthetic Active Radiation**

36) Measurements of PPFD intensity

    OK

37) Removing (per W m-2 nm-1) solves this confusion

    OK

38) • line 486: wavelength ($\lambda_n$)

    Can't trace this in the text

39) See: track changes document: New Eq. 1

    OK

40) See: track changes document: Improved Table 2

    OK

**4 Discussion**

41) Agree, the major factor would be the limited coverage of the PAR band due to narrow band response of the 11 bands.

    Where is this overview given?

**Technical corrections**

42) • line 100: verify autor name. Probably Lopes Pereira.

    Not fully done Peirera => Pereira

---

## Author Response (AR2)

Referee1:

Referee response: L60-61: "no flat" to non-flat. Could just say "and its spectral response is not flat" here and at L56.

Response: Accepted, changed to non-flat, see L55.

Referee response: L139: Rephrase GNSS sensor "was considered" to reflect the fact that this is the solution that was chosen.

Response: Accepted, changed to "was chosen", see L129.
* * *
Referee2:

Abstract

1) See: track changes document: L639-684 OK

2) See: track changes document L417 and L594

Referee response: I have some doubts concerning equation 1 (L417)

- How can a calibration factor be unitless? Or is it just a multiplicative gain-type factor?

Response: It is a multiplier stored in non-volatile read only memory inside each sensor. It normalizes the sensor signal response to a standard as defined by the manufacturer. Since the manufacturer calibrates it for reflection measurements, using 3 different LED's (UV, White and NIR) as an excitation light source (see L267), it is not possible to translate that to a realistic signal response to GHI. We prefer to include the calibration value since it can provide some information about sensor-to-sensor variation, which should be within 15% (according to the manufacturer).

- If Rsensor is the spectral response, it also should have absolute units, otherwise it is just a *relative* spectral response.

Response: It is indeed unitless as it is the normalized peak spectral response. For clarification, we added: "normalized peak" in L395 and we added in L400: "Note that the denominator is the spectrally-weighted source-signal strength.".

- On the form of the equation: if the wavelength dependence of *Rsensor* is expressed - Rsensor_i,$\square$ – so should be the case for *Tdiffuser* and *Tfilter*.

Response: Agreed, we added this in L396, L397

3) See: track changes document: changed to GHI OK

4) See: track changes document: added in line 554 The remark refers to line 19. There is no mention of dark current nor zero offsets at line 554. Please verify back line 19.

**Response: We do refer to and show the zero offsets in Fig. 18 and L409-410. To highlight the zero offsets characteristics we added a note regarding Figs. 13-15 (L485-486).** "Note that in Figs. 13-15, the nocturnal offsets are zero."

5) See: track changes document line 23-24 OK

1. Introduction

6) In line 55 we state: "..., and temperature sensitivity". OK

7) See: track changes document: Michalsky et al, (1991) reference was missing and added in L737-738. They only provide rms errors. OK

8) See: track changes document L56. OK

9) We prefer inserting a reference containing such a Figure. See Fig. 1 of Alados-Arboleda et al., 1995. OK

10) This was developed in lines 73-79 OK

2.1 Light Sensor

11) The more specific term for our sensor would be a "filter-based spectrometer", but it still qualifies as a spectrometer. We will clarify this in line 132. The filters are already described in line 136. See: track changes document: L89: added "filter"

Referee response: In order to qualify as spectrometer, an instrument should measure a wavelength dependent quantity (units ~ W.m-2.nm-1). FROST is measuring integrated signal in 18 different wavebands (units ~ W.m-2). Please note that this is in no way diminishing FROST instrument general quality. It is just a matter of correctness of radiometric definitios. I advise to revise several instances of document in the sense of denominating FROST as a multi-channel radiometer rather than a spectrometer.

**Response: Every spectrometer has a light filtering method, and even our reference spectrometer (ASD FieldSpec) does not have an absolute nm waveband response (it still is a Gaussian distribution). Our calibration is such that its output is W nm$^{-1}$ m$^{-2}$, (see also Fig. 10). Of course,**

we need to understand the waveband response (see Figs. 6 and 8) as is the case for every spectrometer. Thus, our use of "spectrometer" and its output in W $nm^{-1}$ $m^{-2}$ is valid.

12) No, there are no 3 bands, no RGB bands. The Red, Green and Blue are used to identify each of the 3 light detection chips. Each chip detects 6 light wavebands. The manufacturer (AMS) also identifies the 3 chips using the same color coding. We understand that this may sound confusing. Line 135 should clarify this, but based on the reviewer's comment, more info is needed. See: track changes document: color coding improved, see Figs

Referee response: I don't understand if this was corrected as there is no information concerning this on line 135.

Response: There is numerous information regarding the color coding:

-Figure 1 and L105-106.

-Color coding is explained in L141: "We will identify the AS72651, -52, -53 as the blue, red and green sensor, as indicated in Fig. 1."

-Figure 6a, 6b (Blue sensor, Green sensor, Red sensor), L317

-Figure 8, L344-347

-L360-361

-L368-369

-L376

-Table 2

-L441

13) See: track changes document L157 and added +/-10 nm center-wavelength specification OK

14) See: track changes document L669-671 OK

2.1 Diffuser material

15) See: track changes document: L169-170 OK

16) See: track changes document L171-172 OK

3.1 Spectral response and temperature sensitivity

17) See: track changes document: technical specs updated, Fig. 5 caption expanded, added L262-266 OK

18) See: track changes document: Fig. 6a added OK

19) See: track changes document: L273-276 OK

20) We will provide an xy Figure to show the quality of the comparison (supplementary materials)? Unfortunately the experimental data could not be retrieved.

Referee response: OK, but the method should be briefly explained nonetheless

**Response: The method was described in L259-261. For further clarification we added "…in a dark room".**

21) See: track changes document: L279-280 OK

22) See: track changes document: L279-280 OK

23) See: track changes document: some rearrangements and additions: L284-307 OK

24) See: track changes document: improved Figs. 5, 6b, changed: L348-352 OK

25) See: track changes document improved Fig. 8 OK

26) See: track changes document: Line removed and L385 Figure references added OK

27) See: track changes document: L418-436 and new S1. OK

28) See: track changes document: improved Table 2 OK

**3.3 Cosine response and GHI**

29) The FROST cannot decouple diffuse radiation from GHI. OK

30) Agreed, we will add relative units also and improve Figure 13, 14 and 15 captions

Referee response: The error is still shown in absolute units [W.m-2].

Referee response: I think it would be more readable if it is given in percentage.

**Response: Accepted. We included the percentage error above a certain GHI threshold (otherwise we think it makes no sense, i.e. >100% around sunset/sunrise. See L463-464: "Relative errors at GHI >200 W m-2 are <2% and mainly related to horizontal misalignment causing an asymmetric error before/after noon.", L471-472: "Relative errors at GHI >100 W m-2 are <7% and mainly related to spatial separation between FROST and reference." and L479-480:" Relative errors at**

GHI >100 W m-2 are <7% and mainly related to spatial separation between FROST and reference."

31) See: track changes document: All Figs. 13-15 and Fig. captions improved OK

32) See: track changes document: improved Fig. 15 OK

33) See: track changes document: L505-506 OK

34) See: track changes document: L507 added and Figure caption improved
Referee response: OK for changes in Figure 16. But I would still strongly recommend including a mathematical expression.
**Response: Accepted, see Eq. 2.**

**3.4 Spatial measurements and synchronization**
35) We think it is nice to show this as a Figure since it directly visualizes the perfect synchronization, the fast response speed and no zero offset (or dark current) OK

**3.5 Photosynthetic Active Radiation**
36) Measurements of PPFD intensity OK

37) Removing (per W m-2 nm-1) solves this confusion OK

38) • line 486: wavelength ($\boldsymbol{\lambda}$n)

Referee: Can't trace this in the text

**Response: Yes, we remember the typo and it was corrected but we can't trace the exact location.**

39) See: track changes document: New Eq. 1 OK

40) See: track changes document: Improved Table 2 OK

**4 Discussion**
41) Agree, the major factor would be the limited coverage of the PAR band due to narrow band response of the 11 bands.
Referee: Where is this overview given?

Response: We have extensively clarified the wavebands, sensitivity, crosstalk, etc. so one can properly judge the quality for their application. It would be beyond the scope of our manuscript to discuss all limitations for every application.

Technical corrections

42) • line 100: verify autor name. Probably Lopes Pereira. Not fully done Peirera => Pereira

Response: Corrected in L132 and L704.